# Darwinian Optimization: Training Deep Networks with Natural Selection

## Abstract

In conventional deep learning training paradigms, all samples are usually subjected to uniform selective pressure, which fails to adequately account for variations in competitive intensity and diversity among them. This often leads to challenges such as class imbalance bias, insufficient learning of hard samples, and improper handling of noisy samples. Drawing inspiration from the principles of species competition and adaptation in natural ecosystems, we propose a bio-inspired optimization method for deep networks, termed *Natural Selection (NS)*. NS introduces a competition mechanism by first assembling a group of samples into a composite image and then downscaling it to the original input size for model inference. Each sample is then assigned a natural selection score based on the model's predictions on this composite image, reflecting its competitive status within the group. This score is further used to dynamically adjust the loss weight of each sample, facilitating an adaptive network optimization process driven by competitive interactions among training samples. Experimental results on 12 public datasets consistently demonstrate that NS improves performance without being tied to specific network architectures or task assumptions. This study offers a novel perspective on deep network optimization and holds instructive significance for broader applications. The code will be made publicly accessible.

## 1 Introduction

> *"On the origin of species by means of natural selection, or the preservation of favoured races in the struggle for life."* — Charles Robert Darwin

In nature, numerous species coexist within the same ecosystem, each exhibiting genetic diversity within its population while competing for limited resources. According to Darwin's theory of natural selection (Darwin et al., 1859), individuals with greater fitness are more likely to survive, reproduce, and pass on their genes, driving the evolution of species toward higher levels of fitness. Throughout this process, both interspecific and intraspecific competition pressures constantly shape the dynamic patterns of the entire ecosystem.

Analogously, the optimization process of deep networks can be regarded as a mechanism of selection, evolution, and adaptation. Given a dataset with multiple categories, the network updates its parameters continuously during iterative optimization. Each forward and backward pass acts as a *selection* step, where the system computes the loss for each sample and determines parameter updates that minimize the overall training error based on gradients. These updates represent a form of *evolution*, as the network progressively adjusts its parameters to improve performance. Over many iterations, the parameters gradually converge toward an optimal solution that accommodates the majority of samples. This process is similar to how biological populations *adapt* to their environment.

However, classical training methods, such as minimizing the cross-entropy loss with stochastic gradient descent (SGD) (LeCun et al., 2002; Goodfellow et al., 2016), essentially impose a uniform selective pressure on all "species (categories)" and "individuals (samples)" without distinction, failing to adequately account for the differences in competition among samples. In other words, although mainstream network optimization strategies are powerful, they may serve as an inefficient substitute for the complex real evolutionary processes we aim to simulate. This simplification further leads to various issues in network training, such as preference bias caused by class imbalance (Zhang et al.,

2023), inadequate learning of hard samples (Lin et al., 2017), and the misselection of noisy samples (Song et al., 2022). Although strategies such as class reweighting (Cui et al., 2019) and hard example mining (Shrivastava et al., 2016) can alleviate imbalance and hard-sample issues, their heuristic nature demands careful tuning, introduces prior sensitivity, and risks amplifying noise.

To address the above challenges, we propose a biology-inspired method named *Natural Selection (NS)*, which aims to restore an ecosystem-like dynamic balance by explicitly modulating the competitive intensity among training samples. Specifically, we design an explicit competition mechanism: a group of samples is stitched and scaled before being fed into the network for prediction, based on which a natural selection score is computed for each sample. This score reflects the competitive variations among samples and is used to dynamically adjust the training loss of each sample, encouraging more effective and balanced optimization. We validate the effectiveness of the proposed method on four representative image classification tasks. Furthermore, its generalizable nature allows for potential applications to a broader range of domains.

Our main contributions are summarized as follows: 1) We introduce a new perspective inspired by natural selection to advance the research community's understanding of deep network optimization mechanisms. 2) We propose a novel NS method that dynamically adjusts the loss weights of training samples based on their natural selection scores. 3) Extensive experiments on 12 public datasets covering four common computer vision tasks demonstrate the effectiveness of the proposed method.

## 2 RELATED WORK

**Data weighting**. Data weighting methods (Wu & Yao, 2025) can be broadly categorized into two types: class-level weighting and instance-level weighting. Class-level weighting assigns the same weight to all samples within a class, commonly used to alleviate class imbalance issues. Examples include the soft-weighted loss (Zheng et al., 2020) based on class frequency and the class-balanced loss (Cui et al., 2019) derived from the effective number of samples, both of which aim to enhance the model's ability to recognize minority classes. In contrast, instance-level weighting (Kumar et al., 2010; Chang et al., 2017) independently computes or learns a weight for each sample, allowing for a more fine-grained reflection of each sample's importance or learning difficulty during training.

In recent years, instance-level weighting has emerged as the mainstream in data weighting, due to its ability to assign individualized weights to each sample, which shows effectiveness in addressing challenges such as class imbalance (Fernando & Tsokos, 2021; Guo et al., 2022), noisy labels (Liu & Tao, 2015; Jiang et al., 2018; Zhang & Pfister, 2021; Yao et al., 2024), and domain shift (Gu et al., 2021; Xiao & Zhang, 2021). These methods generally fall into several categories: 1) importance-based weighting (Katharopoulos & Fleuret, 2018; Byrd & Lipton, 2019; Fang et al., 2020; Kimura & Hino, 2024; Holstege et al., 2025), which assigns weights by measuring the rarity of a sample within a distribution or its degree of mismatch with the target distribution; 2) meta-learning-based weighting (Ren et al., 2018; Shu et al., 2019; 2023), which formulates weight learning within a bi-level optimization framework, guiding the training of the weighting network by optimizing a meta-loss on a small and clean validation set; 3) difficulty-driven weighting (Lin et al., 2017; Zhou & Wu, 2024), such as focal loss (Lin et al., 2017), which introduces a modulating factor into the cross-entropy loss to suppress gradients from easy samples and amplify learning signals from hard ones. It is worth noting that while most existing methods focus on weighting original training samples, some studies have also explored assigning weights to augmented samples (Yi et al., 2021; Han et al., 2022) . Distinct from the aforementioned approaches, this paper proposes a novel instance-level weighting method inspired by natural selection, which dynamically adjusts the weights of dominant and disadvantaged samples during competition.

**Mix-based data augmentation**. In this work, we utilize image stitching to create synthetic images, a technique that shares some similarities with mix-based data augmentation methods (Cao et al., 2024). As a pioneering data augmentation technique, mixup (Zhang et al., 2018) enhances model generalization by interpolating both images and labels of different samples. Subsequently, researchers have proposed a variety of mixing strategies, such as CutMix (Yun et al., 2019), RI-CAP (Takahashi et al., 2019), RankMixup (Noh et al., 2023), and Similarity Kernel Mixup (Bouniot et al., 2025), which have demonstrated promising effectiveness in improving model robustness and generalization. Although our method also relies on image synthesis, its motivation and objective are fundamentally different from those of the aforementioned data augmentation approaches. Rather

than aiming to increase data diversity, we leverage synthetic images to simulate the mechanism of natural selection, with the goal of adjusting the learning weights assigned to samples with varying competitive advantages, thereby more effectively guiding the model to focus on differences among samples during training.

**Evolutionary computation**. Biology and evolutionary theory have provided substantial inspiration to the field of machine learning, leading to the development of numerous efficient algorithms. Evolutionary computation, as an optimization methodology inspired by natural evolutionary processes, encompasses various classical branches such as genetic algorithms (Holland, 1975; Golberg, 1989) and ant colony optimization (Dorigo, 1992; DRIGO, 1996). Its fundamental idea is to solve complex optimization problems by simulating mechanisms of natural evolution and genetics. In recent years, the integration of evolutionary algorithms with reinforcement learning has become increasingly close, demonstrating strong capabilities in tasks such as policy optimization and environment exploration. Notable progress has been made in areas like neural architecture search (Real et al., 2017; 2019) and evolutionary reinforcement learning (Such et al., 2017; Conti et al., 2018). The method proposed in this paper also draws inspiration from biological mechanisms, aiming to provide a novel bio-inspired solution for optimizing deep neural networks.

## 3 METHODOLOGY

### 3.1 RETHINKING NETWORK OPTIMIZATION THROUGH AN EVOLUTIONARY LENS

The optimization process of deep neural networks exhibits a profound yet often overlooked similarity with the natural evolution of species in ecosystems. For instance, in supervised learning for image classification, the training process of a neural network can be analogized to an evolutionary process within a "digital ecosystem". To formalize this analogy, we first introduce the supervised learning setting as follows:

- Let $\mathcal{X}$ be the input space and $\mathcal{C} = \{1, \ldots, K\}$ represent the set of $K$ discrete class labels. Given a training dataset $\mathcal{D} = \{(x_i, y_i)\}_{i=1}^N$, each $x_i \in \mathcal{X}$ represents an input instance and is paired with a corresponding class label $y_i \in \mathcal{C}$, and $N$ denotes the total number of samples.

- A classifier parameterized by $\theta$ is defined as a function $f_\theta : \mathcal{X} \to \mathbb{R}^K$, which outputs a score or probability vector over the $K$ classes for each input. For a given training sample $(x_i, y_i)$, the loss is computed as $\ell_i(\theta) = \ell(f_\theta(x_i), y_i)$, where $\ell(\cdot, \cdot)$ denotes a chosen loss function, such as the cross-entropy loss.

- The overall training objective is to minimize the empirical risk: $L(\theta) = \frac{1}{N} \sum_{i=1}^N \ell_i(\theta)$. This objective is typically optimized using iterative optimization algorithms, such as stochastic gradient descent (SGD) or Adam.

To establish a mathematical foundation for the evolutionary analogy, we define a mapping between the components of supervised learning and evolutionary concepts:

- **Population structure**: The training dataset $\mathcal{D}$ corresponds to a population of individuals, where each sample $(x_i, y_i)$ represents an individual.

- **Fitness function**: We define the fitness of an individual as $h_i(\theta) = M - \ell_i(\theta)$, where $M$ is a constant ensuring $f_i(\theta) > 0$. This suggests that high fitness is associated with low loss.

- **Selective pressure**: The gradient $\nabla_\theta \ell_i(\theta)$ provides directional pressure analogous to natural selection, driving the system toward fitter states.

**Proposition 1 (Evolutionary-Learning Correspondence).** Under the mapping $\phi$ defined by:

$$\phi(\text{population}) = \mathcal{D}, \qquad \phi(\text{individual } i) = (x_i, y_i),$$
$$\phi(\text{fitness of individual } i) = M - \ell_i(\theta), \quad \phi(\text{selective pressure}) = \nabla_\theta \ell_i(\theta),$$

there exists a structural correspondence between the evolutionary process maximizing population fitness and the learning process minimizing empirical risk, characterized by equivalent optimization objectives and similar iterative improvement dynamics.

We establish the structural correspondence through two key aspects:

1) **Objective function duality**. The evolutionary objective of maximizing average fitness:

$$\max_{\theta} \frac{1}{N} \sum_{i=1}^{N} h_i(\theta) = \max_{\theta} \frac{1}{N} \sum_{i=1}^{N} [M - \ell_i(\theta)] \tag{1}$$

is mathematically equivalent to the learning objective of minimizing empirical risk:

$$\min_{\theta} \frac{1}{N} \sum_{i=1}^{N} \ell_i(\theta) \tag{2}$$

since maximizing $M - \ell_i(\theta)$ is equivalent to minimizing $\ell_i(\theta)$ for constant $M$.

2) **Improvement dynamics**. Both processes employ iterative improvement strategies: In evolution, fitter individuals are more likely to reproduce, gradually improving population fitness; In learning, parameters are updated in the direction that reduces loss: $\theta_{t+1} = \theta_t - \eta \cdot \nabla_\theta L(\theta)$. While their update process differ, both mechanisms drive the system toward better performance over time.

This formal correspondence provides a mathematical foundation for interpreting network optimization through an evolutionary lens, highlighting the structural similarities between biological evolution and learning optimization.

By establishing an analogy between network optimization and ecosystem evolution, we can critically re-examine the limitations of the classical supervised learning paradigm with Empirical Risk Minimization (ERM). The standard ERM framework minimizes the average loss over the training dataset: $\min_\theta \frac{1}{N} \sum_{i=1}^{N} \ell(x_i, y_i; \theta)$, which implicitly assigns equal weight $w_i = 1$ to every sample $(x_i, y_i)$. This uniform weighting scheme, while statistically well-motivated, fails to account for the competitive dynamics among samples. In ecological terms, the ERM objective applies a constant selective pressure $\nabla_\theta \ell(x_i, y_i; \theta)$ to all individuals, regardless of their fitness or the population composition. The gradient signal that drives optimization, $\frac{1}{N} \sum_{i=1}^{N} \nabla_\theta \ell(x_i, y_i; \theta)$, represents a global average that lacks the fine-grained adaptation to individual competitive advantages.

The core limitation lies in the homogeneity of the weighting function $w_i \equiv 1$, which prevents the emergence of dynamic selection pressures analogous to those in natural ecosystems. Consequently, the network optimization process cannot prioritize samples with higher fitness or protect developing minority patterns $\mathcal{D}_{\text{min}}$ from being overwhelmed by dominant patterns $\mathcal{D}_{\text{maj}}$. From an evolutionary perspective, this eliminates the natural selection mechanism essential for maintaining diversity and adapting to complex environments, often leading to premature convergence and limited generalization capability.

In essence, by neglecting individual-level competition mechanisms akin to natural selection, classical training methods are thus more prone to issues such as local optima, overfitting, or underfitting. *In paradigms lacking such competition, can Darwinian natural selection be incorporated into the training process to restructure the optimization dynamics?*

## 3.2 NATURAL SELECTION OF TRAINING SAMPLES

Before addressing the above question, we first revisit the mechanism of natural selection in nature. As shown in Fig. 1 (a), natural selection is primarily driven by two-fold competition: 1) interspecific competition, in which species compete for limited resources, and 2) intraspecific competition, where individuals within a species compete for survival and reproduction. The core of both types of competition lies in the competition among individuals. Inspired by this mechanism, we introduce similar competitive relations into network training: samples compete for update resources according to their current competitive status and are subjected to varying degrees of selective pressure, thereby constructing an ecological dynamic balance during training.

To simulate individual competition and evaluate their relative advantages, we propose a method termed *Natural Selection (NS)*, which assigns each sample an NS score via explicit inter-sample competition. As illustrated in Fig. 1 (b), given a set of training samples $\mathcal{G} = \{(x_i, y_i)\}_{i=1}^{m}$ (e.g., $m = 4$), where each sample has a spatial size of $H_0 \times W_0 \times 3$, we first arrange the samples into an $R \times C$ grid layout (with $R \times C = m$) and concatenate them along spatial dimensions to form a composite image:

$$S = \text{Stitch}(x_1, x_2, \ldots, x_m) \in \mathbb{R}^{R \cdot H_0 \times C \cdot W_0 \times 3}. \tag{3}$$

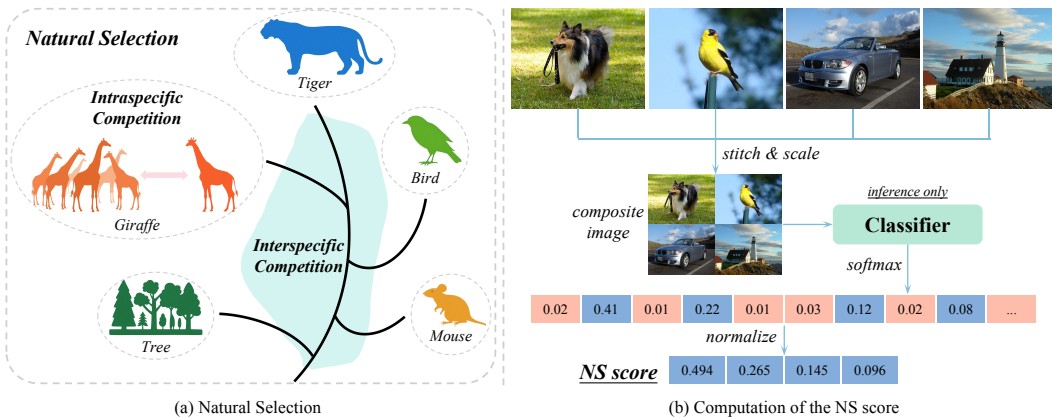

(a) Natural Selection          (b) Computation of the NS score

Figure 1: (a) Natural selection in ecosystem evolution. (b) Illustration of the natural selection process in training samples. After performing image stitching and scale normalization, the resulting stitched image is fed into the classifier to obtain the softmax probability distribution. The predicted probability associated with each sample's category is then extracted, and the NS score is subsequently computed through normalization.

The composite image is subsequently rescaled to the original input size via bilinear interpolation, producing $S' \in \mathbb{R}^{H_0 \times W_0 \times 3}$, which is then passed to the classifier $f_\theta(\cdot)$ to obtain the output logits $z = f_\theta(S') \in \mathbb{R}^K$. Notably, $f_\theta(\cdot)$ is used solely for forward inference (excluded from backpropagation) and isolated from the standard training pipeline, thus exerting no influence on model optimization. Then, the class-posterior probability (confidence) vector is obtained via $p = \text{softmax}(z) \in [0,1]^K$, with $\sum_{i=1}^{K} p_i = 1$.

For sample $x_i$, we take the probability of its ground-truth class as the raw competitive score $q_i = p[y_i]$. To induce explicit competition within the group and reduce cross-group scale discrepancies, we normalize these scores to obtain the NS score:

$$s_i = \frac{q_i}{\sum_{j=1}^{m} q_j}. \tag{4}$$

During this process, an explicit competition occurs: a sample $x_i$ with a larger $s_i$ indicates a stronger competitive advantage (the winner), while a smaller $s_i$ indicates a weaker advantage (the loser). The vector $s = (s_1, \ldots, s_m)$ quantifies inter-sample competitiveness and serves as the basis for applying selective pressure during network optimization. The PyTorch implementation for computing the NS score is provided in Appendix A.

### 3.3 OPTIMIZING DEEP NETWORKS VIA NATURAL SELECTION

After obtaining the NS score, how to leverage it to optimize deep networks becomes a critical issue. An intuitive approach is to use the NS score as a sample weight to modulate its loss, thereby introducing differentiated selective pressure. However, this leads to a fundamental dilemma regarding how the weighting should be allocated: *Should we strengthen the winners or focus more on the losers?* Prioritizing the winners would mean optimizing those that have already demonstrated stronger competitive advantages, whereas emphasizing the losers would imply assigning greater importance to samples that are in an unfavorable position during competition throughout optimization.

To address this issue, we propose the following two weighting strategies based on the NS score, each tailored to different learning scenarios:

1) **NS-based Winner-Strengthening** (NS-WS) strategy assigns larger weights to samples with higher NS scores, aiming to consolidate samples that already exhibit stronger competitive advantages. This strategy enhances the stability of gradient signals, promotes the sharpening of decision boundaries, and facilitates the convergence of optimization. Specifically, when the NS score effectively reflects sample learnability or label reliability, emphasizing high-score samples helps reduce disturbances caused by low-quality samples, making training more stable and efficient. NS-WS is

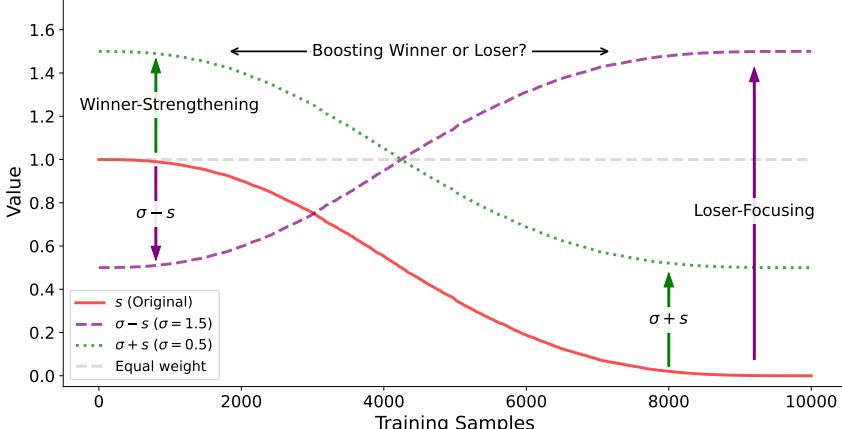

Figure 2: The distribution curves of sample weights under different settings. Under the NS-WS (green dashed line) and NS-LF (purple dashed line) strategies, samples with higher-weight values are reinforced during training, while those in lower-weight regions are suppressed. This enables differentiated optimization for samples with varying competitive characteristics.

particularly suitable for scenarios requiring high discriminability and stable decision boundaries. In tasks involving label noise or annotation ambiguity, it can still be effective provided that a positive correlation exists between NS scores and sample reliability, as it suppresses the adverse impact of potentially mislabeled samples on the optimization trajectory.

2) **NS-based Loser-Focusing** (NS-LF) strategy assigns greater weights to samples with lower NS scores to enhance their influence on parameter updates. Intuitively, samples with low NS scores are disadvantaged in the current competition, which may correspond to instances from long-tail categories, boundary cases, or examples that have not yet been sufficiently learned. By increasing the relative weight of such samples, the model allocates more capacity during training to rectify these under-optimized areas, thereby improving performance on minority classes and challenging regions, and enhancing overall generalization and robustness. This strategy is suitable for scenarios that require fairness or long-tail adaptation, such as tasks with imbalanced class distributions, where boundary samples significantly affect performance, or during early training phases where broader exploration is desired to avoid over-reliance on easy examples. Importantly, the effectiveness of NS-LF depends on the assumption that the NS score reflects samples that are learnable yet currently suppressed rather than those dominated by noise or annotation errors.

It should be noted that both strategies entail potential risks. NS-WS may overly rely on samples with high NS scores, resulting in selection bias that overlooks disadvantaged samples and minority classes, thereby limiting generalization. Conversely, NS-LF could amplify the influence of noisy and mislabeled samples, leading to training instability. To mitigate these issues, we introduce a base value $\sigma$ to adjust the numerical range of NS mapping and provide a lower bound for the weight of each sample. Formally, the final per-sample weight is defined as:

$$w_i = \sigma + \rho \cdot s_i, \tag{5}$$

where $\rho \in \{+1, -1\}$ denotes the strategy polarity: Setting $\rho = +1$ yields the NS-WS strategy, while choosing $\rho = -1$ implements to the NS-LF strategy.

As illustrated in Fig. 2, we compare the weight distribution curves simulated over 10,000 samples under three settings: 1) The red solid curve represents the original mapping in descending order. In this case, the weight of samples with low NS scores $s_i$ approaches zero, resulting in negligible gradient contributions during training and thus limiting the model's ability to learn from these examples. 2) The green dashed curve corresponds to the NS-WS strategy ($\rho = +1$) with an additive base value $\sigma$. This term preserves the relative ordering among samples while effectively smoothing the weight distribution, preventing individual samples from being assigned near-zero weights or allowing extremely high-score samples to dominate the update. 3) The purple dashed curve denotes the NS-LF strategy ($\rho = -1$), which globally increases the relative weight of low-score samples and

suppresses the influence of high-score ones, while $\sigma$ ensures that the weights remain within a stable range and avoid extreme values, thereby promoting training stability.

After obtaining the sample weights from Eq. (5), we apply them directly in the per-sample training loss, defining the weighted loss as $\ell_i(\theta) = w_i \cdot \ell(f_\theta(x_i), y_i)$. Since the calculation of the NS score is independent of the training process, this mechanism can be seamlessly integrated into existing training frameworks to enable competition-aware dynamic optimization of deep networks.

## 4 EXPERIMENTS

### 4.1 DATASETS AND EXPERIMENTAL DETAILS

**Datasets**. The proposed method is comprehensively evaluated on a total of 12 public datasets across four mainstream computer vision tasks, including *image classification* (**CIFAR-10** (Krizhevsky, 2009), **CIFAR-100** (Krizhevsky, 2009), **ImageNet-1K** (Russakovsky et al., 2015)), *emotion recognition* (**Twitter I** (You et al., 2015), **Twitter II** (Borth et al., 2013), **Flickr** (Katsurai & Satoh, 2016), **Instagram** (Katsurai & Satoh, 2016), **FI** (You et al., 2016), **EmoSet** (Yang et al., 2023)), *source-free domain adaptation* (**Office-Home** (Venkateswara et al., 2017)), and *long-tailed classification* (**CIFAR-LT-10** (Cui et al., 2019), **CIFAR-LT-100** (Cui et al., 2019)). Detailed information about these datasets can be found in Appendix B.

**Experimental details**. To ensure consistency with existing research, we conduct evaluations using widely adopted backbone networks across multiple datasets and tasks. The selected backbones cover convolutional neural networks such as AlexNet (Krizhevsky et al., 2012), VGGNet (Simonyan & Zisserman, 2015), ResNet (He et al., 2016), WideResNet (Zagoruyko & Komodakis, 2016), ResNeXt (Xie et al., 2017), and DenseNet (Huang et al., 2017), transformer architectures including ViT (Dosovitskiy et al., 2020) and Swin Transformer (Liu et al., 2021), as well as the emerging Vision Mamba (VMamba) (Liu et al., 2024) architecture. For reproducible and comparable findings, all experiments are repeated three times with final metrics derived from mean values across repetitions. Further details are provided in Appendix C.

**Competitors**. We choose 12 representative methods for comparison, i.e., Focal Loss (Lin et al., 2017), GCE (Zhang & Sabuncu, 2018), label smoothing (LS) (Müller et al., 2019), NLNL (Kim et al., 2019), SCE (Wang et al., 2019), APL (Ma et al., 2020), LOW (Santiago et al., 2021), PolyLoss (Leng et al., 2022), ANL (Ye et al., 2023), AUL (Zhou et al., 2023), CE$_\epsilon$+MAE (Wang et al., 2024), CE+OGC (Ye et al., 2025). These methods are devised from different perspectives and optimize the training process by improving the sample loss. In addition, we select three classical sampling-based methods for comparison, i.e., Class-Balanced Sampling (CBS), Square-Root Sampling (SRS), and Progressively-Balanced Sampling (PBS), with details provided in Appendix D.

Table 1: Evaluation of NS on **ImageNet-1K**. "WS" denotes boosting the winner, while "LF" means enhancing the loser. "2/4" indicates grouping two or four samples for competition.

| Method | Basic | w/ NS | | | |
|---|---|---|---|---|---|
| | | LF-2 | WS-2 | LF-4 | WS-4 |
| ResNet-18 | 68.4 | 68.4 | 68.6 | 68.4 | **68.9** (↑0.5) |
| Swin-T | 75.2 | 75.4 | 75.5 | 75.2 | **75.7** (↑0.5) |
| VMamba-T | 76.7 | 76.9 | 76.9 | 76.8 | **77.1** (↑0.4) |

Table 2: Evaluation of NS-WS on **CIFAR-10/100** with different networks.

| Method | CIFAR-10 | | CIFAR-100 | |
|---|---|---|---|---|
| | Basic | w/ NS-WS | Basic | w/ NS-WS |
| AlexNet | 77.4 | **78.6** (↑1.2) | 43.8 | **46.0** (↑2.2) |
| VGG-19 | 93.4 | **93.8** (↑0.4) | 71.7 | **72.6** (↑0.9) |
| ResNet-110 | 92.8 | **93.9** (↑1.1) | 70.3 | **72.1** (↑1.8) |
| ResNeXt-29 | 95.9 | **96.2** (↑0.3) | 81.4 | **82.3** (↑0.9) |
| WRN-28-10 | 96.1 | **96.4** (↑0.3) | 81.1 | **81.5** (↑0.4) |

### 4.2 MAIN RESULTS

**Image classification**. We evaluate the proposed NS method on the large-scale ImageNet-1K dataset (Russakovsky et al., 2015) using three different types of networks. Specifically, we compare four experimental settings that combine two group sizes (2 and 4 samples per group) with two enhancement strategies (i.e., NS-WS and NS-LF). Results are summarized in Table 1. We empirically observe that boosting winners improves classification accuracy, whereas enhancing losers yields no significant

Table 3: Comparison of basic and NS-enhanced models on six emotion recognition datasets.

| Method | Twitter I | | Twitter II | | Flickr | | Instagram | | FI | | EmoSet | |
|---|---|---|---|---|---|---|---|---|---|---|---|---|
| | Basic | w/ NS-LF | Basic | w/ NS-LF | Basic | w/ NS-LF | Basic | w/ NS-LF | Basic | w/ NS-LF | Basic | w/ NS-LF |
| AlexNet | 73.8 | **75.2** | 68.5 | **71.0** | 83.2 | **83.8** | 80.6 | **81.2** | 56.4 | **57.9** | 67.6 | **69.2** |
| VGG-16 | 75.3 | **78.1** | 74.3 | **74.9** | 84.4 | **84.6** | 82.5 | **83.1** | 59.7 | **61.4** | 72.2 | **73.9** |
| ResNet-50 | 81.3 | **82.9** | 71.0 | **74.3** | 85.7 | **86.4** | 84.6 | **85.2** | 65.8 | **67.2** | 75.8 | **76.3** |
| DenseNet-121 | 78.9 | **79.4** | 72.8 | **74.0** | 85.1 | **85.6** | 84.0 | **85.0** | 63.2 | **64.4** | 74.9 | **75.2** |
| ViT-B/16 | 74.5 | **76.4** | 72.1 | **72.2** | 86.2 | **86.4** | 84.5 | **85.3** | 65.6 | **67.4** | 77.3 | **77.7** |
| Swin-T | 69.3 | **69.7** | 70.0 | **73.1** | 83.9 | **85.9** | 82.3 | **82.5** | 64.4 | **67.0** | 76.9 | **78.4** |
| Avg. | 75.5 | **77.0** | 71.5 | **73.3** | 84.8 | **85.5** | 83.1 | **83.7** | 62.5 | **64.2** | 74.1 | **75.1** |

Table 4: Comparison of different methods on the **Office-Home** benchmark.

| Method | Ar→Cl | Ar→Pr | Ar→Rw | Cl→Ar | Cl→Pr | Cl→Rw | Pr→Ar | Pr→Cl | Pr→Rw | Rw→Ar | Rw→Cl | Rw→Pr | Avg. |
|---|---|---|---|---|---|---|---|---|---|---|---|---|---|
| Baseline | 58.0 | 78.8 | 81.9 | 69.4 | 79.7 | 78.7 | 68.2 | 55.0 | 81.9 | 73.8 | 58.7 | 83.8 | 72.3 |
| Focal Loss | 57.9 | 78.8 | 82.0 | 69.2 | 79.8 | 78.6 | 68.3 | 55.4 | 81.8 | 73.7 | 59.9 | 83.7 | 72.4 |
| GCE | 57.5 | 77.7 | 80.9 | 68.8 | 76.5 | 77.3 | 66.9 | 54.5 | 80.9 | 73.8 | 58.7 | 84.1 | 71.5 |
| LS | 58.0 | 78.7 | 82.0 | 69.3 | 79.7 | 78.6 | 68.3 | 55.3 | 81.9 | 73.8 | 59.7 | 83.9 | 72.4 |
| NLNL | 56.4 | 77.5 | 80.1 | 67.4 | 75.6 | 76.3 | 65.5 | 54.8 | 80.5 | 73.4 | 59.0 | 83.5 | 70.8 |
| SCE | 54.5 | 78.3 | 80.6 | 63.8 | 76.7 | 76.2 | 65.3 | 50.4 | 80.6 | 70.5 | 53.7 | 82.8 | 69.4 |
| APL | 57.2 | 77.8 | 82.1 | 68.9 | 76.8 | 78.5 | 67.7 | 54.1 | 81.5 | 73.1 | 58.4 | 84.1 | 71.7 |
| LOW | 58.1 | 78.8 | 82.1 | 69.5 | 79.7 | 78.5 | 68.2 | 55.3 | 81.9 | 73.8 | 59.5 | 83.9 | 72.4 |
| PolyLoss | 57.9 | 78.6 | 82.0 | 69.7 | 79.7 | 78.5 | 67.9 | 55.0 | 81.9 | 73.5 | 59.6 | 83.9 | 72.3 |
| ANL | 57.1 | 77.6 | 80.3 | 67.5 | 75.6 | 76.4 | 65.7 | 54.9 | 80.6 | 73.4 | 59.2 | 83.8 | 71.0 |
| AUL | 56.0 | 77.9 | 81.6 | 66.6 | 79.0 | 77.7 | 66.0 | 52.4 | 81.1 | 72.6 | 57.5 | 84.0 | 71.0 |
| $CE_\epsilon$+MAE | 56.9 | 77.9 | 80.6 | 68.4 | 76.0 | 76.9 | 66.4 | 54.4 | 80.8 | 73.6 | 59.2 | 84.0 | 71.3 |
| CE+OGC | 58.6 | 79.0 | 82.1 | 69.9 | 79.1 | 78.6 | 68.6 | 55.0 | 81.6 | 73.8 | 60.0 | 84.1 | 72.5 |
| *Sampling-based methods* | | | | | | | | | | | | | |
| CBS | 57.9 | 78.3 | 81.7 | 68.7 | 79.6 | 78.3 | 68.3 | 55.4 | 82.1 | 72.8 | 59.4 | 84.2 | 72.2 |
| SRS | 57.4 | 78.7 | 81.6 | 68.4 | 79.5 | 78.5 | 68.0 | 56.1 | 81.9 | 73.5 | 58.4 | 84.5 | 72.2 |
| PBS | 57.6 | 78.5 | 82.2 | 68.4 | 79.7 | 78.6 | 68.4 | 56.3 | 82.2 | 73.5 | 59.5 | 83.9 | 72.4 |
| **NS-WS (ours)** | **59.1** | **79.1** | **82.4** | **70.0** | **79.8** | **78.8** | **68.8** | **57.0** | **82.3** | **73.9** | **60.1** | **85.0** | **73.0** |

improvement. This finding contradicts the conventional intuition that disadvantaged samples should be prioritized for enhancement. However, upon further analysis, we find this behavior reasonable for datasets like ImageNet-1K, which have relatively balanced class distributions but non-negligible label noise. Enhancing winners helps the model learn more generalizable class-specific features while mitigating the negative impact of noisy samples. In addition, results show that grouping four samples per competition achieves superior performance compared to grouping two. Overall, the experimental results demonstrate that NS-WS can consistently improve performance without relying on specific network architectures.

We adopt the same "WS-4" configuration on the standard CIFAR10 and CIFAR100 datasets (Krizhevsky, 2009) and evaluate the proposed method using five widely-used networks, with results shown in Table 2. The results show that across different networks on both datasets, NS-WS achieves an accuracy improvement of 0.3% to 2.2%, further confirming its generalization effectiveness in image classification tasks.

**Emotion recognition**. Given the varying degrees of class imbalance in these datasets, we adopt the WS-LF strategy to give greater attention to samples at a competitive disadvantage. We conduct experiments on six commonly used emotion recognition datasets and select six classic deep networks as baseline models to evaluate the effect of introducing the NS-LF strategy. As shown in Table 3, different networks exhibit noticeable performance differences across datasets. NS-LF does not rely on a specific network architecture and consistently improves recognition accuracy, with average gains ranging from 0.6% to 1.8%, which demonstrates its effectiveness for emotion recognition.

### 4.3 COMPARATIVE RESULTS

**Source-free domain adaptation (SFDA)**. Given the significant pseudo-label noise issue in SFDA tasks, we employ the NS-WS strategy to suppress the impact of noisy samples. In the experiments,

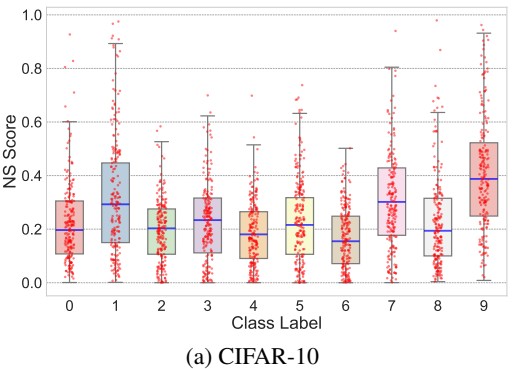 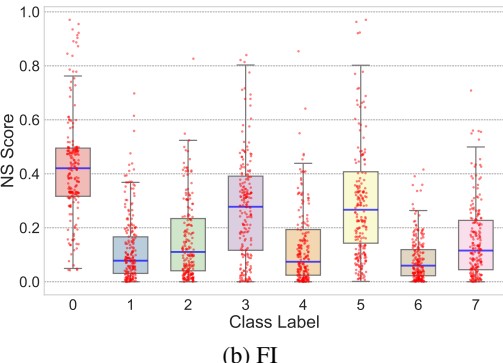

(a) CIFAR-10                    (b) FI

Figure 3: Visualization of NS score distributions by category on the CIFAR-10 and FI datasets. Red points denote training samples, and the blue solid line within each box indicates the median NS score of all samples in that class.

we use SHOT (Liang et al., 2020) as the baseline and compare NS-WS with existing methods across all 12 sub-tasks of Office-Home (Venkateswara et al., 2017). As shown in Table 4, the majority of these existing methods provide minimal or no performance improvement, with several leading to pronounced performance drops in the SFDA task. In contrast, NS-WS increases the average accuracy by 0.7% and outperforms all competitors on every sub-task. Since SFDA settings often involve noisy pseudo-labels, these results highlight the superior effectiveness of NS-WS for network optimization in such challenging scenarios.

**Long-tailed classification**. For this typical class-imbalanced task, the NS-LF strategy is employed to enhance the model's learning capability for minority class samples. We employ ResNet-32 (He et al., 2016) as the baseline and conduct comparative experiments on the CIFAR-LT-10 and CIFAR-LT-100 datasets (Cui et al., 2019). Each dataset includes four imbalance factor settings (200, 100, 50, 10), representing the ratio between the number of samples in the most frequent and the least frequent classes. The results in Table 5 show that as the imbalance ratio increases from 10 to 200, the accuracy of all methods exhibits a declining trend. Compared to the baseline, a few methods, such as Focal Loss and LOW, achieve improvements under some imbalance factors, though the gains are limited, and most methods show negligible or even degraded performance. In contrast, our NS-LF delivers consistent improvements across all imbalance factor settings, demonstrating its effectiveness in optimizing deep networks under long-tailed data distributions.

Table 5: Performance comparison on the **CIFAR-LT-10/100** datasets.

| Method | CIFAR-LT-10 | | | | CIFAR-LT-100 | | | |
|---|---|---|---|---|---|---|---|---|
| | 200 | 100 | 50 | 10 | 200 | 100 | 50 | 10 |
| Baseline | 64.0 | 69.5 | 72.7 | 86.7 | 33.6 | 38.4 | 42.1 | 53.8 |
| Focal Loss | 60.2 | 69.5 | 73.4 | 86.9 | 33.6 | 37.6 | 43.6 | 55.4 |
| GCE | 46.3 | 60.3 | 63.8 | 86.2 | 18.6 | 21.8 | 25.3 | 36.8 |
| LS | 62.5 | 70.3 | 74.8 | 86.8 | 32.9 | 37.8 | 42.9 | 56.0 |
| NLNL | 33.0 | 35.1 | 39.0 | 52.6 | 1.9 | 1.6 | 1.6 | 2.2 |
| SCE | 59.3 | 66.2 | 74.4 | 10.0 | 1.0 | 31.0 | 36.3 | 53.6 |
| APL | 29.9 | 30.8 | 31.3 | 52.6 | 6.9 | 9.2 | 11.3 | 15.1 |
| LOW | 63.0 | 70.3 | 74.4 | 87.0 | 33.2 | 38.9 | 43.1 | 56.2 |
| PolyLoss | 59.4 | 69.7 | 74.5 | 86.6 | 34.2 | 39.6 | 42.9 | 56.0 |
| ANL | 53.1 | 67.5 | 74.8 | 85.3 | 15.7 | 17.2 | 18.6 | 28.7 |
| AUL | 28.4 | 28.4 | 35.4 | 45.4 | 3.7 | 3.6 | 5.2 | 4.5 |
| $CE_\epsilon$+MAE | 53.4 | 61.4 | 65.2 | 86.1 | 26.0 | 30.2 | 34.9 | 48.8 |
| CE+OGC | 42.0 | 43.9 | 44.2 | 77.1 | 24.3 | 27.2 | 36.3 | 45.9 |
| *Sampling-based methods* | | | | | | | | |
| CBS | 61.0 | 68.5 | 75.1 | 86.6 | 26.8 | 30.5 | 38.5 | 53.9 |
| SRS | 61.6 | 69.9 | 75.2 | 86.5 | 32.2 | 37.4 | 41.6 | 55.1 |
| PBS | 63.6 | 68.7 | 75.2 | 86.8 | 28.7 | 31.4 | 38.3 | 54.6 |
| **NS-LF (ours)** | **65.9** | **71.2** | **75.3** | **87.3** | **35.4** | **39.7** | **43.9** | **56.4** |

## 4.4 FURTHER ANALYSIS

**NS score distribution**. On the CIFAR-10 dataset, we employ the ResNet-110 model, while on the FI dataset, we utilize the ResNet-50 model to statistically analyze the distribution of NS scores across categories (taking one training epoch as an example), as shown in Fig. 3. To illustrate competitive divergence among samples more clearly, we randomly select 200 samples from each category for visualization. The box plots in Figs. 3 (a) and (b) reveal marked differences in NS score distributions across categories, reflecting competitive relationships between classes. Meanwhile, the scatter plots of samples within each category reveal notable variation in NS scores, even

among samples from the same class, demonstrating intra-class competition. These results suggest that competitive differences among samples, as measured by NS scores, effectively mirror the interspecific and intraspecific competition in ecosystems. Consequently, the NS score serves as an effective metric for quantifying sample competitiveness and establishes a basis for incorporating a natural selection-inspired dynamic equilibrium into the network optimization process.

**Correlation analysis**. We conduct class-wise analyses on the CIFAR-LT-10/100 datasets under class-imbalance settings. This allows us to examine the correlation between class sample size and mean NS score. On the Office-Home dataset under the SFDA setting, we examine the correlation between class accuracy and mean NS score. As shown in Figs. 4 (a) and (b), the mean NS score exhibits a significant positive correlation with class sample size. This indicates that classes with more samples tend to have higher average NS scores and thus a relative competitive advantage. Furthermore, Fig. 4 (c) shows a positive correlation between the mean NS score and class accuracy. In the SFDA scenario, this typically means that classes with higher pseudo-label quality (lower noise) also have higher NS scores. These observations provide empirical support for NS score-guided optimization: moderately enhancing competitively disadvantaged samples in class-imbalanced environments, and prioritizing competitively advantaged samples to provide relatively reliable supervision in SFDA. We emphasize that these findings are based on correlational rather than causal inference, and the effectiveness of our method has been validated through the aforementioned experiments. See Appendix G for more results (Figs. 5 and 6).

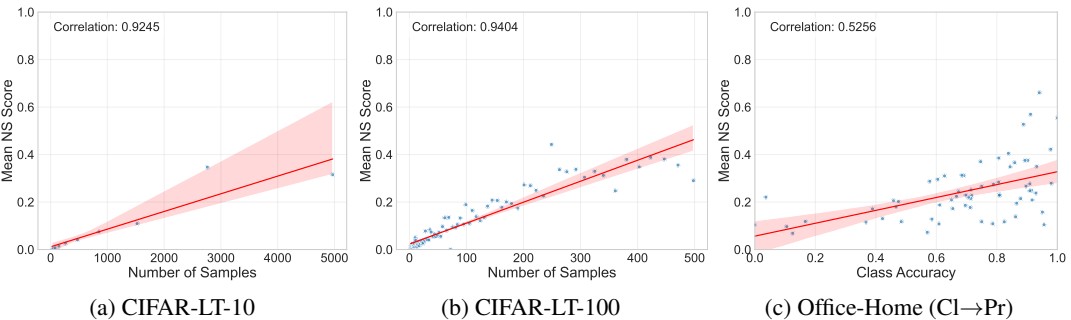

|                |                  |                        |
| :------------: | :--------------: | :--------------------: |
| (a) CIFAR-LT-10 | (b) CIFAR-LT-100 | (c) Office-Home (Cl→Pr) |

Figure 4: Correlation analysis on the CIFAR-LT-10/100 and Office-Home datasets. Blue points represent the individual categories, while the red line depicts the fitted linear regression.

## 5 CONCLUSION

Traditional deep network training methods typically apply uniform selective pressure to each sample, ignoring inherent competitive differences among them. Inspired by biological mechanisms, this paper proposes a natural selection method that explicitly simulates competitive interactions between samples for ecologically balanced optimization. Specifically, we stitch a group of images for joint prediction, evaluating the competitive strength of each sample to compute a corresponding natural selection score. This score then dynamically modulates each sample's training loss, thereby enhancing overall network optimization. Experimental results across 12 public datasets covering four types of computer vision tasks demonstrate that the proposed method achieves consistent performance gains without relying on specific network architectures. We emphasize that the NS method offers notable usability and generalization potential. Its applicability is not limited to the tasks validated in this paper and shows promise for extension to a broader range of research areas.

## ETHICS STATEMENT

We confirm that we have read and strictly adhered to the ICLR ethical guidelines. This research focuses on general deep learning optimization techniques and does not involve human subjects. We are committed to conducting responsible research and hope that the proposed technology can be applied to a broader range of AI fields. All experiments are based on publicly available datasets, and we have made every effort to ensure the reproducibility and fairness of the study.

## REPRODUCIBILITY STATEMENT

To ensure the reproducibility of this work, we have taken the following measures: the core code of the proposed NS method has been released in Appendix A, and the experimental configurations and implementation details are elaborated in Appendix C. Furthermore, all experiments were conducted on publicly available benchmark datasets. We hope these resources will be sufficient to assist researchers in reproducing our results.

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

# A  ALGORITHM

The Natural Selection algorithm is easy to implement. The following code implements the NS score computation in PyTorch. This code stitches input images in groups of 4, with each sample having dimensions of $3 \times 224 \times 224$ (channel $\times$ height $\times$ width).

```python
import torch
import torch.nn as nn
import torch.nn.functional as F

def natural_selection(inputs, targets, classifier, B):
    """
    inputs: training samples, [B, 3, 224, 224].
    targets: category labels.
    classifier: deep network.
    B: batch size.
    """
    group_num = int(B / 4)
    grouped = inputs.view(group_num, 4, 3, 224, 224)

    # Stitch and scale
    top = torch.cat([grouped[:, 0, :, :, :], grouped[:, 1, :, :, :]], dim=2)
    bottom = torch.cat([grouped[:, 2, :, :, :], grouped[:, 3, :, :, :]], dim=2)
    mix_image = torch.cat([top, bottom], dim=3)
    downsampled = F.interpolate(
        mix_image,
        size=(224, 224),
        mode='bilinear',
        align_corners=False
    )

    # Predict and normalize
    mix_scores = nn.Softmax(dim=1)(classifier(downsampled))
    row_indices = torch.arange(group_num).unsqueeze(1).expand(-1, 4)
    result = mix_scores[row_indices, targets.view(group_num, 4)]
    ns_score = (result / result.sum(dim=1, keepdim=True)).view(-1)

    return ns_score
```

# B  DATASETS

In this section, we provide a brief introduction to the datasets used in our experiments.

## B.1  IMAGE CLASSIFICATION

For the standard image classification task, we select two well-established datasets (CIFAR-10/100) and the large-scale ImageNet-1K for evaluation.

- **CIFAR-10** (Krizhevsky, 2009) is a highly classic image dataset comprising 10 common object categories. It contains a total of 60,000 small-sized images, each with a resolution of $32 \times 32$ pixels.
- **CIFAR-100** (Krizhevsky, 2009) is an expanded and more complex benchmark derived from the CIFAR-10 dataset. It comprises 60,000 $32 \times 32$ pixel color images, equally distributed across 100 classes.
- **ImageNet-1K** (Russakovsky et al., 2015) is the most widely used subset of the full ImageNet database, often serving as the standard benchmark for image classification. It comprises approximately 1.28 million high-resolution training images and 50,000 validation images, meticulously annotated across 1,000 object classes.

## B.2 EMOTION RECOGNITION

For visual emotion recognition, we evaluate our approach on six publicly available datasets. Following Zheng et al. (2025), we use refined versions of these datasets that exclude problematic images, such as duplicates and corrupt files, to improve the reliability of performance evaluation.

- **Twitter I** (You et al., 2015) is a small-scale sentiment classification dataset constructed from images on the Twitter platform. It contains two sentiment labels, positive and negative, with a total of 1,251 images.
- **Twitter II** (Borth et al., 2013) shares a similar structure with Twitter I, also containing two sentiment categories, but is smaller in scale, consisting of only 545 images.
- **Flickr** (Katsurai & Satoh, 2016) is also a two-class dataset, but on a significantly larger scale, comprising a total of 42,848 images collected from the Flickr platform.
- **Instagram** (Katsurai & Satoh, 2016) is a two-class sentiment dataset collected from the Instagram platform, which is also larger in scale, with a total of 60,729 images.
- **FI** (You et al., 2016) covers eight emotion categories: amusement, awe, contentment, excitement, anger, disgust, fear, and sadness. All images in this dataset are collected from the Flickr and Instagram platforms, with a total of 21,829 images.
- **EmoSet** (Yang et al., 2023) is a large-scale dataset for emotion recognition that contains the same eight emotion categories as the FI dataset, with a total of 118,102 images.

## B.3 SOURCE-FREE DOMAIN ADAPTATION

For source-free domain adaptation, we employ the widely used **Office-Home** (Venkateswara et al., 2017) benchmark for evaluation, which consists of images from four domains: Artistic images (Ar), Clip art (Cl), Product images (Pr), and Real-world images (Rw). It includes 15,500 images across 65 categories of everyday objects.

## B.4 LONG-TAILED CLASSIFICATION

For the long-tailed image classification task, we adopt two widely used datasets for validation.

- **CIFAR-LT-10** (Cui et al., 2019) is a long-tailed version derived from the classic CIFAR-10 dataset. It preserves the original 10 classes while artificially inducing an exponentially decreasing number of samples per class in the training set according to a predefined imbalance factor. The test set remains the same as in the original dataset.
- **CIFAR-LT-100** (Cui et al., 2019) is a long-tailed dataset constructed based on CIFAR-100, retaining all 100 categories. Its training set is generated using the same construction method as CIFAR-LT-10, while the test set remains in its original balanced state.

# C EXPERIMENTAL DETAILS

This section introduces the details of experiments conducted on various datasets using different network architectures across four computer vision tasks. Unless otherwise specified, all experiments are implemented in PyTorch and run on a single NVIDIA RTX 4090 GPU.

## C.1 IMAGE CLASSIFICATION

On the CIFAR-10 and CIFAR-100 datasets, we select five commonly used deep networks for evaluation, i.e., AlexNet (Krizhevsky et al., 2012), VGG-19 (Simonyan & Zisserman, 2015), ResNet-110 (He et al., 2016), ResNeXt-29 (Xie et al., 2017), and WRN-28-10 (Zagoruyko & Komodakis, 2016). We optimize using SGD with a momentum of 0.9, a batch size of 128, and an initial learning rate of 0.1. All models share this optimizer configuration, while training schedules and learning rate decay differ by architecture: AlexNet, VGG-19, and ResNet-110 are trained for 164 epochs with learning rate multiplied by 0.1 at epochs 81 and 122; ResNeXt-29 is trained for 300 epochs with decay by a

factor of 0.1 at epochs 150 and 225; WRN-28-10 is trained for 200 epochs with decay by a factor of 0.2 at epochs 60, 120, and 160.

On the ImageNet-1K dataset, we choose ResNet-18 (He et al., 2016), Swin-T (Liu et al., 2021), and VMamba-T (Liu et al., 2024) as representative baseline networks, and train all models for 90 epochs. For ResNet-18, we use the SGD optimizer with momentum 0.9, set the batch size to 256, and adopt an initial learning rate of 0.1. We decay the learning rate by a factor of 0.1 at epochs 31 and 61. For Swin-T and VMamba-T, we train with four NVIDIA RTX 4090 GPUs using the AdamW optimizer (Kingma & Ba, 2015) with a weight decay of 0.05, employ a cosine learning rate scheduler with five epochs of linear warmup, set a global batch size of 1024 (i.e., 256 per GPU), and use an initial learning rate of 0.001.

For the proposed NS method, we employ the NS-WS strategy across all three datasets and compute the NS score using groups of four samples. The base value hyperparameter $\sigma$ was set to 0.7, 0.8, and 2.0 for CIFAR-10, CIFAR-100, and ImageNet-1K, respectively.

## C.2 EMOTION RECOGNITION

We select six classic deep networks as baselines, i.e., AlexNet (Krizhevsky et al., 2012), VGG-16 (Simonyan & Zisserman, 2015), ResNet-50 (He et al., 2016), DenseNet-121 (Huang et al., 2017), ViT-B/16 (Dosovitskiy et al., 2020), and Swin-T (Liu et al., 2021). All models are trained using SGD with momentum 0.9, a batch size of 64, an initial learning rate of 1e-3, and a polynomial learning rate decay. They are fine-tuned for 10 epochs based on ImageNet-pretrained weights. Due to class imbalance in emotion recognition datasets, we apply the NS-LF strategy across all datasets. On Twitter I and Twitter II, the NS scores are computed with groups of two samples, and the base value $\sigma$ is set to 2.5. On Flickr and Instagram, we use groups of four samples with a base value $\sigma$ of 2.0. On FI and EmoSet, we use groups of four samples with a base value $\sigma$ of 2.5.

## C.3 SOURCE-FREE DOMAIN ADAPTATION

Following SHOT (Liang et al., 2020), we use ResNet-50 (He et al., 2016) as the feature extraction backbone and adopt a classifier consisting sequentially of a bottleneck layer (fully connected layer followed by batch normalization) and a weight-normalized fully connected layer. For optimization, we employ SGD with a momentum of 0.9 and a batch size of 64. The learning rate is set to 1e-3 for the backbone and 1e-2 for the bottleneck. The final fully connected classification layer, serving as the source hypothesis, is frozen during target adaptation. Before adaptation, we first pre-train the source model on the source domain in a supervised manner. During target domain adaptation, all models are trained for 15 epochs. In our proposed NS method, the hyperparameter $\sigma$ is set to 0.6, and the NS scores are computed over groups of four samples. Considering the pseudo-label noise present in SFDA, we adopt the NS-WS strategy to improve model robustness.

## C.4 LONG-TAILED CLASSIFICATION

We implement our method using SGD with momentum 0.9 and a batch size of 128. We adopt ResNet-32 (He et al., 2016) as the backbone network, and train the model for 240 epochs. The initial learning rate is set to 0.1 and decayed at epochs 100, 160, and 200 by factors of 0.1, 0.1, and 0.01, respectively. Given the inherent class imbalance in long-tailed classification, we apply the NS-LF strategy on both CIFAR-LT-10 and CIFAR-LT-100 datasets to enhance the model's ability to recognize minority-class samples. For NS score computation, the hyperparameter $\sigma$ is set to 2.5 with groups of 2 samples on CIFAR-LT-10, and to 1.5 with groups of 4 samples on CIFAR-LT-100.

# D DETAILS OF SAMPLING-BASED METHODS

**Class-Balanced Sampling (CBS)**. This method aims to make the expected sampling probability equal across classes. A typical implementation uses a two-stage strategy: first uniformly sampling a class from the class set, and then uniformly selecting a sample from that class. Equivalently, it can be viewed as assigning a sampling weight to each sample from class $k$ that is inversely proportional to its class frequency $n_k$.

**Square-Root Sampling (SRS)**. As a common variant of class-balanced sampling, this method applies a square-root transformation to class frequencies to smooth sampling bias. Concretely, the class-level sampling probability is proportional to $\sqrt{n_k}$, and with the two-stage implementation, the instance-level probability becomes proportional to $\frac{1}{\sqrt{n_k}}$. This achieves a trade-off between frequency-sensitive and class-balanced sampling.

**Progressively-Balanced Sampling (PBS)**. This method performs dynamic interpolation between frequency-based sampling and class-balanced sampling. The sampling probability is a weighted combination of them, with the weight of class-balanced sampling gradually increasing during the training process to achieve a smooth transition from frequency-preference to class balance.

## E  STATISTICAL RESULTS

In our experiments, we perform multiple runs using three random seeds (2024, 2025, 2026) and report the average performance. We systematically compile the mean results and standard deviations across the 12 public datasets considered in this study, as shown in Table 6. Overall, the performance gains achieved by our method are significantly greater than random fluctuations, indicating that the observed improvements are statistically significant.

Table 6: Statistical results across the 12 public datasets.

| Datasets | mean±std | Datasets | mean±std | Datasets | mean±std | Datasets | mean±std |
|---|---|---|---|---|---|---|---|
| ImageNet-1K | 73.9±0.05 | Twitter I | 77.0±0.45 | Instagram | 83.7±0.21 | Office-Home | 73.0±0.19 |
| CIFAR-10 | 91.8±0.08 | Twitter II | 73.3±0.34 | FI | 64.2±0.26 | CIFAR-LT-10 | 74.9±0.43 |
| CIFAR-100 | 70.9±0.22 | Flickr | 85.5±0.11 | EmoSet | 75.1±0.12 | CIFAR-LT-100 | 43.9±0.18 |

## F  ABLATION STUDY

We conduct an experimental analysis on the CIFAR-100 dataset using ResNet-110, focusing on the image stitching grouping strategies as well as parameters $\sigma$ and $\rho$ involved in the proposed method.

1) We evaluate five different grouped stitching configurations, including 1×2 (2 images), 2×2 (4 images), 2×4 (8 images), 4×2 (8 images), and 4×4 (16 images) layouts. As shown in Table 7, our evaluation reveals that group sizes of 2 and 4 both achieve better and comparable performance. Therefore, we adopt the 2/4-group configuration across all experiments.

Table 7: Evaluation of five different grouped stitching configurations.

| Group | 1×2 | 2×2 | 2×4 | 4×2 | 4×4 |
|---|---|---|---|---|---|
| Acc (%) | 72.0 | **72.1** | 71.7 | 71.4 | 71.7 |

2) We evaluate the base value $\sigma$ in Eq. (5). As shown in Table 8, setting $\sigma = 0$ leads to significant performance degradation, primarily because the original NS scores of some samples are too small (close to zero), which hinders effective network optimization. The model achieves optimal performance when $\sigma$ is set to 0.8. Given that our study involves multiple tasks with substantial differences in dataset distributions, we recommend adopting a similar hyperparameter selection strategy to determine the optimal $\sigma$ value for different applications.

Table 8: Performance with different values of $\sigma$.

| $\sigma$ | 0.0 | 0.1 | 0.5 | 0.8 | 1.0 | 1.5 | 1.8 |
|---|---|---|---|---|---|---|---|
| Acc (%) | 67.5 | **70.9** | 71.3 | **72.1** | 71.7 | 71.5 | 71.4 |

3) On the CIFAR-100 dataset, we adopt the NS-based winner-strengthening strategy with the parameter $\rho$ set to 1. To validate the rationality of this parameter choice, we further evaluate the model

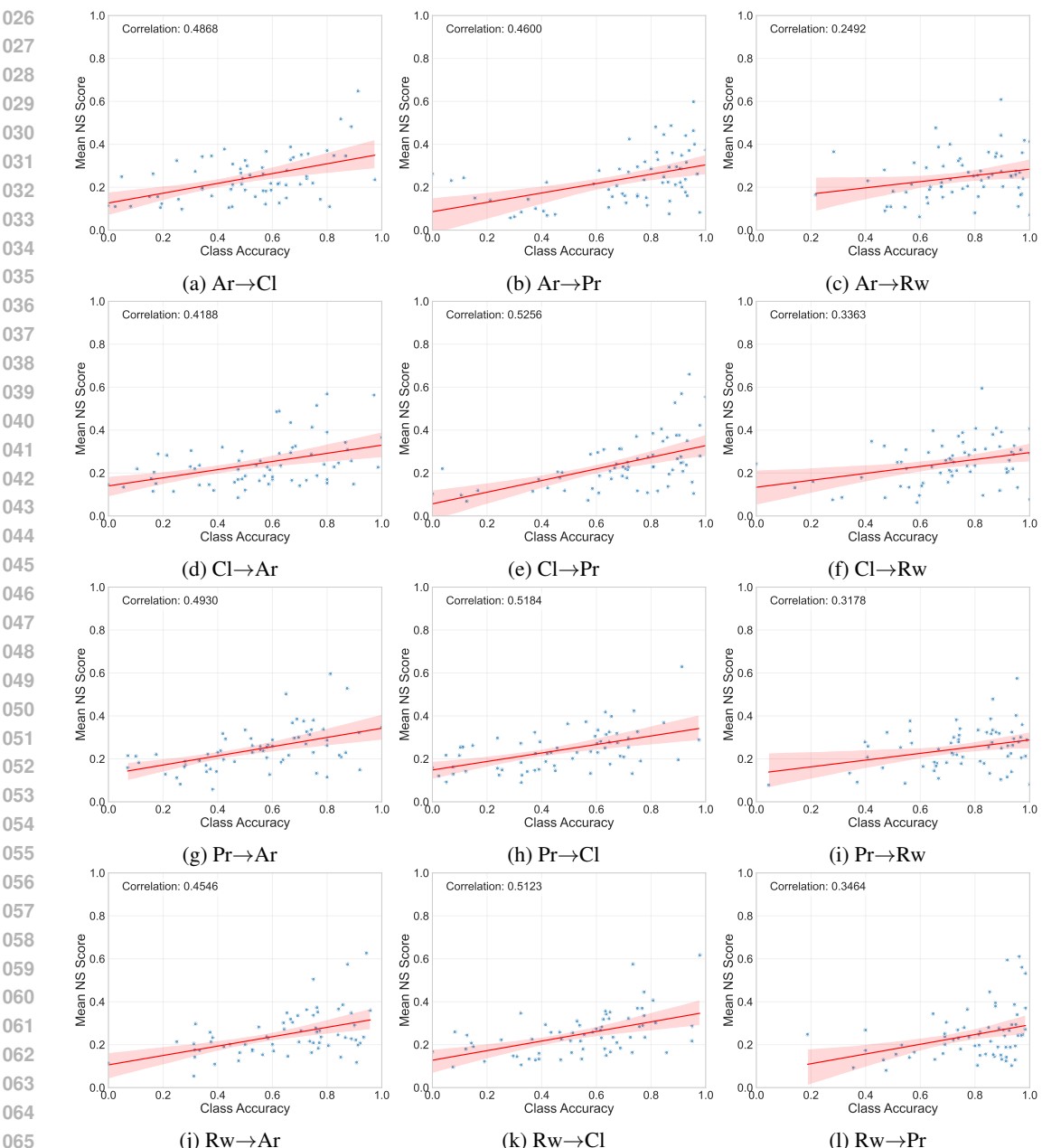

Figure 5: Correlation between class accuracy and mean NS scores on **Office-Home**.

performance with different values of $\rho$. As shown in Table 9, the model achieves optimal performance when $\rho = 1$. Based on this empirical result, we set $\rho$ at 1 in our experiments without further fine-tuning of this parameter.

Table 9: Performance with different values of $\rho$.

| $\rho$ | 0.0 | 0.1 | 0.5 | 0.8 | 1.0 | 1.5 | 1.8 |
|---|---|---|---|---|---|---|---|
| Acc (%) | 70.3 | **71.3** | 71.6 | 71.8 | **72.1** | 71.9 | 71.8 |

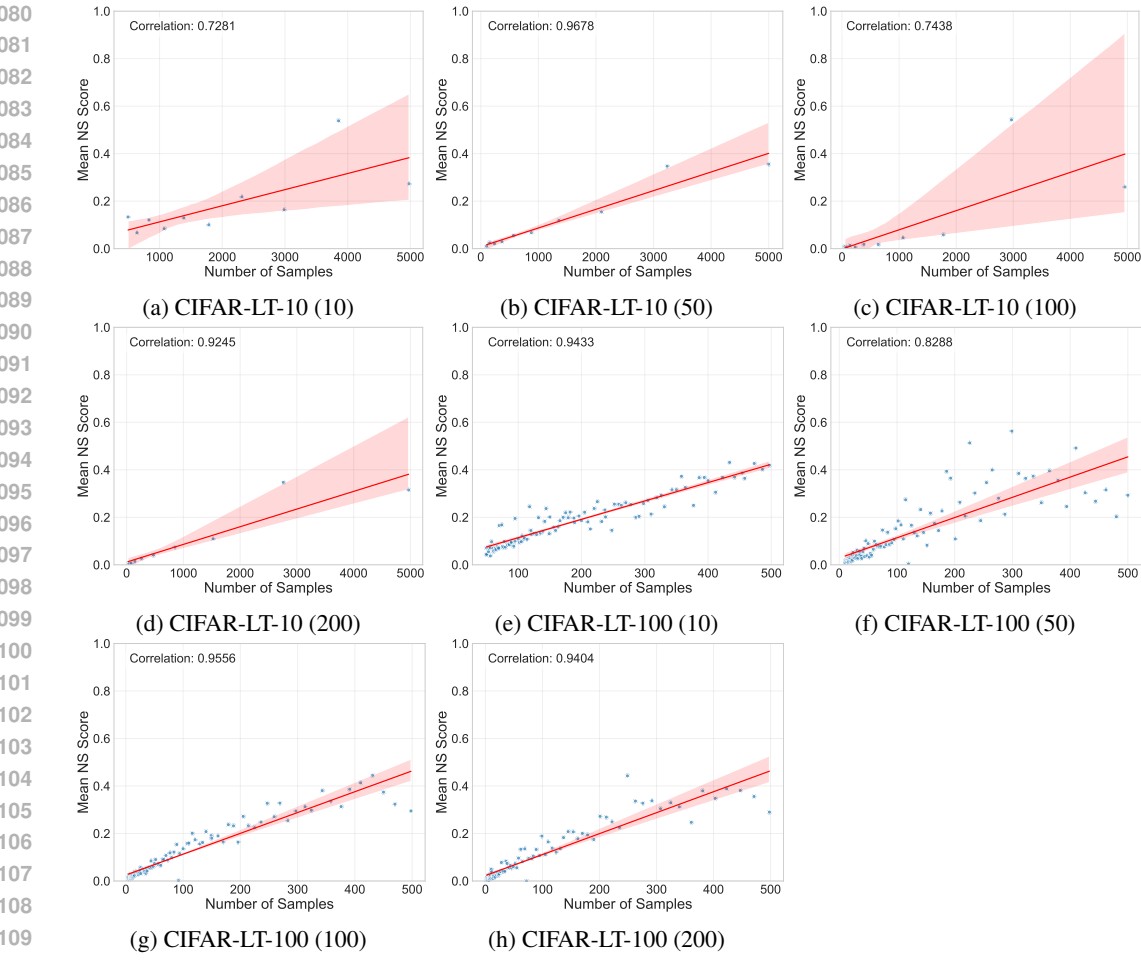

Figure 6: Correlation between number of samples and mean NS scores on **CIFAR-LT-10/100**. The numbers in (·) represent the imbalance factor.

## G   CORRELATION ANALYSIS

Fig. 5 illustrates the correlation between per-class accuracy and mean NS score across all 12 source→target adaptation tasks in the Office-Home dataset, visualized via scatter plots and linear regression fits. Figs. 6 (a)–(d) show the class-wise relationship between class sample size and mean NS score on CIFAR-LT-10 under four different imbalance factors, while Figs. 6 (e)–(h) report the corresponding results on CIFAR-LT-100 under the same four imbalance configurations. All figures are visualized at the category level to facilitate cross-category comparison and trend analysis.

## H   ADDITIONAL RESULTS

To examine whether the NS-WS strategy adversely affects the handling of hard samples, we conduct a dedicated evaluation on the ImageNet-Hard dataset (Taesiri et al., 2023). This dataset comprises numerous images that are generally challenging for existing models to classify correctly, making it suitable for evaluating the robustness of methods on hard samples. Using ResNet-18 as the baseline, we perform experiments as summarized in Table 10. It can be observed that after incorporating the NS-WS strategy, the model's performance on hard samples do not exhibit significant degradation. This indicates that the proposed method poses a low risk of marginalizing legitimate hard samples or minority patterns.

Table 10: Results on the **ImageNet-Hard** dataset.

| Dataset | ResNet-18 | w/ NS-WS |
|---|---|---|
| ImageNet-Hard | 9.9 | **10.2** |

Furthermore, we conduct additional experiments on text classification tasks, using two public datasets (IMDB (Maas et al., 2011) and AG News (Zhang et al., 2015)) and employing TextCNN (Kim, 2014) and BERT-base (Devlin et al., 2019) as baseline models for evaluation. Specifically, for text-based samples, we concatenate four text instances into a single input, which is then converted into sequence or token representations. Subsequently, we compute the NS score for each text via model inference and apply the NS-LF strategy to adjust the training loss accordingly. The experimental results presented in the Table 11 demonstrate that our method achieves consistent performance improvements in text classification tasks, indicating its cross-modal applicability.

Table 11: Results on the **ImageNet-Hard** dataset.

| Dataset | TextCNN | w/ NS-LF | BERT-base | w/ NS-LF |
|---|---|---|---|---|
| IMDB | 75.6 | **76.6** | 91.8 | **92.2** |
| AG News | 90.3 | **91.0** | 94.4 | **94.7** |

## I    COMPUTATION COST

We evaluated the computational overhead introduced by the proposed NS method on the CIFAR-10 and CIFAR-100 datasets using the ResNet-110 network, with a primary focus on GPU memory usage and training time. The results are shown in Table 12. In terms of memory consumption, although the number of classes differs between the two datasets, leading to a variation in the number of parameters in the final layer, this difference is negligible due to the minimal proportion of these parameters in the overall model. Experiments show that the baseline memory usage remains consistent across both datasets. On this basis, NS introduces only a 1.5% additional memory overhead, demonstrating high memory efficiency. Regarding training time, we report the average results over 10 independent runs. Compared to the baseline, the introduction of NS increases training time by 10.2% and 12.7% on CIFAR-10 and CIFAR-100, respectively. This overhead comes mainly from the image stitching, scaling, and model inference in the NS module. The timing analysis for each step in the NS computation process is presented in Table 13. The results demonstrate that the additional computational cost primarily stems from the model inference stage, while the time overhead introduced by image stitching and scaling operations is negligible. Still, the increase remains within an acceptable range. It does not form a significant computational bottleneck.

In summary, as a plug-and-play functional module, NS achieves performance improvements while introducing only negligible memory overhead and limited additional training time, demonstrating strong practical utility.

Table 12: Comparative analysis of GPU memory and per-epoch training time.

| Method | CIFAR-10 | | CIFAR-100 | |
|---|---|---|---|---|
| | GPU memory (MB) | Training time (s) | GPU memory (MB) | Training time (s) |
| ResNet-110 | 1,564 | 23.81 | 1,564 | 23.93 |
| w/ NS | 1,588 (↑1.5%) | 26.24 (↑10.2%) | 1,588 (↑1.5%) | 26.97 (↑12.7%) |

## J    FUTURE WORK

In this paper, we demonstrate the effectiveness of the Natural Selection (NS) method across multiple computer vision tasks. As a general machine learning approach, NS exhibits significant potential for addressing a broader spectrum of learning challenges. Its applications extend to diverse domains,

Table 13: Per-step runtime analysis of NS.

| Step | CIFAR-10 | CIFAR-100 |
|---|---|---|
| Image stitching and scaling | 0.14 s | 0.14 s |
| Model inference | 2.31 s | 2.87 s |

including fine-grained visual classification, few-shot learning, video-based action recognition, and vision-based robotic manipulation tasks. Furthermore, the principles underlying NS can inspire the design of novel bio-inspired network architectures and new paradigms for optimizing the learning frameworks of multimodal large language models in future research.

## K  THE USE OF LARGE LANGUAGE MODELS (LLMS)

In preparing this manuscript, large language models are primarily used for editorial tasks, such as grammar checks, wording refinements, and sentence-level modifications. They are not utilized for content generation, method design, data analysis, or the formation of results and conclusions. All modifications are reviewed and approved individually by the authors to ensure academic accuracy.

