# OpenReview forum: "Darwinian Optimization: Training Deep Networks with Natural Selection"
_ICLR.cc/2026/Conference — Submitted to ICLR 2026_

### Official Review · Reviewer_Bfq8 · 2025-10-30

**Soundness:** 3
**Presentation:** 4
**Contribution:** 4
**Rating:** 8
**Confidence:** 4

**Summary:**

Quite interesting work; A novel Darwinian perspective to optimization dynamics in NN. The paper presents a novel bio-inspired optimization method called Natural Selection (NS) that introduces explicit competition among training samples. By computing competitive scores through image stitching and dynamically adjusting sample loss weights, the method as empirically shown achieves consistent improvements across diverse computer vision tasks.

**Strengths:**

I quite enjoyed reading this paper, which offers a refreshing biology-inspired perspective on optimization for deep networks.

1) Novel research problem: The authors adequately point out the uniform handling of samples in current optimization theory and raise an intuitive question about whether introducing an explicit competition mechanism could be beneficial.
2) Clear visualization: Figure 1 provides an effective visual explanation of their core idea, making the concept accessible.
3) Well-articulated analogy: The authors provide an easy-to-understand mapping between network optimization and ecosystem evolution, along with a succinct research question that is well-formulated in Section 3.1.
4) Thorough empirical validation: The experimental evaluation is comprehensive across multiple tasks. I especially appreciate the thoroughness of the analysis, for example in the long-tailed classification experiments, which demonstrates the method's effectiveness across diverse scenarios (especially following the observations in line 329-339).

**Weaknesses:**

Weaknesses
1) Computational overhead: 10-13% training time increase (Table 6) is non-trivial for large-scale applications.
2) Group size limitations: Why groups of 2 or 4? The authors should include some comment/explanation or intuition around this or include a systematic exploration of other group sizes.
3) Hyper-parameter sensitivity: The base value $\sigma$ seems to vary significantly across tasks (0.6 to 2.5) as per Appendix, suggesting the method may require careful tuning per dataset. Is there a way to tackle this?
4) I think naively combining images through simple concatenation may introduce artificial edge patterns that don't reflect natural competition. This may potentially cause the NS scores to reflect stitching artifacts rather than genuine sample competitiveness. I think that could explain some of the task-dependent behavior and hyper-parameter sensitivity observed across experiments? Do they authors saw/infer something similar as well?

While I don't think this is a weakness - I do think that the authors should comment briefly on this in the paper: Could a variant of NS that does backpropagate through the competition mechanism learn better competitive evaluations over time? Or would this introduce instability and other issues?

**Questions:**

1) Do the authors have some idea how can they tackle the computation overhead limitation?
2) How is the optimal group size determined across different tasks?Is group size a sensitive hyper-parameter? What principles guide the selection—does it depend on batch size, number of classes, or dataset characteristics?
3) How does NS behave in the presence of outliers or anomalous samples? Based on the image classification results (Section 4.2), the Winner-Strengthening strategy performs better on ImageNet, which has label noise. Does favoring winners risk marginalizing legitimate hard samples or minority patterns that appear outlier-like?
4) Do the authors believe this approach will transfer effectively to text-based or other non-vision tasks? The stitching mechanism is naturally suited to images, but how would competition be simulated for text (concatenation of sentences/tokens?)

---

> ### Author Response · Authors · 2025-11-27
> **Official Comment by Authors (Part 1)**
>
> We sincerely thank you for your encouraging feedback and valuable suggestions. We have thoroughly addressed each of your comments by adding new experiments and analysis. Your insights have greatly strengthened the paper, for which we are truly grateful.
>
> **W1. Computational overhead: 10-13% training time increase (Table 6) is non-trivial for large-scale applications.**
>
> **Q1. Do the authors have some idea how can they tackle the computation overhead limitation?**
>
> Thank you for the constructive question. The primary computational overhead of our method stems from the calculation of NS scores. In our implementation, this score is computed in each iteration to ensure timeliness. We observe that model parameters generally do not change significantly between adjacent training epochs, which allows the use of a low-frequency triggering mechanism to reduce overall computation. Specifically, the NS score is calculated only once every k epochs, which theoretically reduces the additional computational overhead to 1/k.
>
> To validate the effectiveness of this strategy, we conducted comparative experiments on CIFAR-10 and CIFAR-100 using ResNet-110, evaluating model performance and time cost under different acceleration ratios. As shown in the table below, increasing the value of k leads to a substantial reduction in training time, while the corresponding degradation in accuracy remains marginal. The results demonstrate that adopting a low-frequency triggering strategy (e.g., with k = 4) preserves the majority of performance gains while significantly reducing additional computational overhead.
>
> &nbsp;&nbsp;&nbsp;&nbsp;&nbsp;&nbsp;&nbsp;&nbsp;&nbsp;&nbsp;&nbsp;&nbsp;&nbsp;&nbsp;&nbsp;&nbsp;&nbsp;&nbsp;&nbsp;&nbsp;&nbsp;&nbsp;&nbsp;&nbsp;&nbsp;&nbsp;&nbsp;&nbsp; **CIFAR-10**&nbsp;&nbsp;&nbsp;&nbsp;&nbsp;&nbsp;&nbsp;&nbsp;&nbsp;&nbsp;&nbsp;&nbsp;&nbsp;&nbsp;&nbsp;&nbsp;&nbsp;&nbsp;&nbsp;&nbsp;&nbsp;&nbsp;&nbsp;&nbsp;&nbsp;&nbsp;&nbsp;&nbsp;&nbsp;&nbsp;&nbsp;&nbsp;&nbsp;&nbsp;&nbsp;**CIFAR-100**
>
> |Method|Acc (%)|Training time (s)|Acc (%)|Training time (s)|
> |:-|:-:|:-:|:-:|:-:|
> |Baseline|92.8|23.81|70.3|23.93|
> |NS-1/1|93.9 (↑1.1)|26.24 (↑10.2%)|72.1 (↑1.8)|26.97(↑12.7%)|
> |NS-1/2|93.7 (↑0.9)|25.04   (↑5.2%)|72.1 (↑1.8)|25.52  (↑6.6%)|
> |NS-1/3|93.7 (↑0.9)|24.57   (↑3.2%)|72.0 (↑1.7)|24.94  (↑4.2%)|
> |NS-1/4|93.6 (↑0.8)|24.38   (↑2.4%)|72.0 (↑1.7)|24.83  (↑3.8%)|
> |NS-1/5|93.5 (↑0.7)|24.28  (↑2.0%)|71.9 (↑1.6)|24.68  (↑3.1%)|
> |NS-1/6|93.5 (↑0.7)|24.17   (↑1.5%)|71.8 (↑1.5)|24.59   (↑2.8%)|
> |NS-1/7|93.4 (↑0.6)|24.12   (↑1.3%)|71.7 (↑1.4)|24.44 (↑2.1%)|
> |NS-1/8|93.4 (↑0.6)|24.08   (↑1.1%)|71.6 (↑1.3)|24.36   (↑1.8%)|
> |NS-1/9|93.4 (↑0.6)|24.06   (↑1.0%)|71.6 (↑1.3)|24.30   (↑1.5%)|
> |NS-1/10|93.2 (↑0.4)|24.05   (↑1.0%)|71.5 (↑1.2)|24.27  (↑1.4%)|

---

> > ### Author Response · Authors · 2025-11-27
> > **Official Comment by Authors (Part 3)**
> >
> > **W4. I think naively combining images through simple concatenation may introduce artificial edge patterns that don't reflect natural competition. This may potentially cause the NS scores to reflect stitching artifacts rather than genuine sample competitiveness. I think that could explain some of the task-dependent behavior and hyper-parameter sensitivity observed across experiments? Do they authors saw/infer something similar as well?**
> >
> > Thank you for raising this important question. To investigate whether the NS score computed based on stitched images might be influenced by edge artifacts, we conducted additional experiments across 12 subtasks of the Office-Home dataset. Specifically, we filled the 24-pixel-wide region along the stitching seam with mean pixel values to suppress potential artifacts near the stitching boundary. We then recalculated the NS score on the processed images, thus minimizing the impact of edge regions on the score.
> >
> > To quantitatively compare the correlation between the original NS score and the score obtained after masking the edges, we employed both the Pearson Correlation Coefficient (PCC) and Spearman's Rank-Order Correlation Coefficient (SROCC). These metrics assess the linear and monotonic relationships between the two sets of scores, respectively. As shown in the table below, both PCC and SROCC values were very high, indicating strong agreement between the NS scores calculated before and after edge masking. This result suggests that the NS score is largely unaffected by artificial edge patterns and is primarily determined by the semantic content within the image.
> >
> > | Statistical Measure | Ar→Cl  | Ar→Pr  | Ar→Rw  | Cl→Ar  | Cl→Pr  | Cl→Rw  | Pr→Ar  | Pr→Cl  | Pr→Rw  | Rw→Ar  | Rw→Cl  | Rw→Pr  |
> > |:------------------- |:------:|:------:|:------:|:------:|:------:|:------:|:------:|:------:|:------:|:------:|:------:|:------:|
> > | PCC                 | 0.9084 | 0.9109 | 0.9319 | 0.9230 | 0.8814 | 0.9177 | 0.9336 | 0.9058 | 0.9321 | 0.9276 | 0.9099 | 0.9025 |
> > | SROCC               | 0.9180 | 0.9236 | 0.9396 | 0.9258 | 0.8919 | 0.9238 | 0.9344 | 0.9132 | 0.9408 | 0.9314 | 0.9178 | 0.9119 |
> >
> > **W5. While I don't think this is a weakness - I do think that the authors should comment briefly on this in the paper: Could a variant of NS that does backpropagate through the competition mechanism learn better competitive evaluations over time? Or would this introduce instability and other issues?**
> >
> > Thank you for the thoughtful question. To investigate this, we conducted additional experiments on CIFAR-10 and CIFAR-100 using ResNet-110. In these experiments, we integrated the NS score calculation into the backpropagation process (denoted as NS-WS*). As shown in the table below, this setting results in a noticeable performance drop, indicating that it is detrimental to network learning. We attribute this primarily to the substantial distribution shift between stitched composite and real images. Integrating the NS score computation into backpropagation can introduce training instability, which consequently degrades model performance.
> >
> > | Dataset   | ResNet-110 | w/ NS-WS | w/ NS-WS* |
> > |:--------- |:----------:|:--------:|:---------:|
> > | CIFAR-10  |    92.8    | **93.9** |   92.1    |
> > | CIFAR-100 |    70.3    | **72.1** |   67.3    |

---

> ### Author Response · Authors · 2025-11-27
> **Official Comment by Authors (Part 2)**
>
> **W2. Group size limitations: Why groups of 2 or 4? The authors should include some comment/explanation or intuition around this or include a systematic exploration of other group sizes.**
>
> **Q2. How is the optimal group size determined across different tasks?Is group size a sensitive hyper-parameter? What principles guide the selection—does it depend on batch size, number of classes, or dataset characteristics?**
>
> Thank you for your attention to the group size. The choice of group size is guided by two considerations. First, since the training batch size is typically set to a power of two, such as 64 or 128, we set the group size to values like 2, 4, 8, and 16 to enable complete grouping without remainder. Second, when the group size exceeds 4, the resulting composite images inevitably contain sub-images positioned at the edges or center, which may introduce undesirable positional bias.
>
> To systematically evaluate the impact of group size, we compared five different grouping configurations on the CIFAR-100 dataset using ResNet-110. The layouts tested were 1×2 (2 images), 2×2 (4 images), 2×4 (8 images), 4×2 (8 images), and 4×4 (16 images). The results are presented in the table below. We observed that group sizes of 2 and 4 both yielded good and comparable performance, indicating that this hyperparameter (2 or 4) is relatively insensitive for this task.
>
> | Group   | 1x2  |   2x2    | 2x4  | 4x2  | 4x4  |
> |:------- |:----:|:--------:|:----:|:----:|:----:|
> | Acc (%) | 72.0 | **72.1** | 71.7 | 71.4 | 71.7 |
>
> Furthermore, we analyzed the mean NS score for each spatial position within the 4 × 4 composite images. The results below revealed that samples located in the central-upper regions of the composite image had higher mean NS scores, while those positioned at the edges, particularly the lower edges, exhibited lower mean scores. This finding confirms that the tiling layout introduces a systematic positional bias, which adversely affects fair competition among samples. Based on this analysis, we adopt group sizes of 2 or 4 in our experiments to reduce the impact of such bias.
>
> &nbsp;&nbsp;&nbsp;&nbsp;&nbsp;&nbsp;&nbsp;&nbsp;&nbsp;&nbsp;1&nbsp;&nbsp;&nbsp;&nbsp;&nbsp;&nbsp;&nbsp;&nbsp;&nbsp;&nbsp;&nbsp;&nbsp;&nbsp;2&nbsp;&nbsp;&nbsp;&nbsp;&nbsp;&nbsp;&nbsp;&nbsp;&nbsp;&nbsp;&nbsp;3&nbsp;&nbsp;&nbsp;&nbsp;&nbsp;&nbsp;&nbsp;&nbsp;&nbsp;&nbsp;&nbsp;&nbsp;4
>
> 1&nbsp;&nbsp;&nbsp;0.0618&nbsp;&nbsp;**0.0645**&nbsp;&nbsp;**0.0644**&nbsp;&nbsp;0.0633
>
> 2&nbsp;&nbsp;&nbsp;0.0614&nbsp;&nbsp;**0.0649**&nbsp;&nbsp;**0.0640**&nbsp;&nbsp;0.0627
>
> 3&nbsp;&nbsp;&nbsp;0.0612&nbsp;&nbsp;0.0626&nbsp;&nbsp;0.0626&nbsp;&nbsp;0.0619
>
> 4&nbsp;&nbsp;&nbsp;0.0612&nbsp;&nbsp;0.0613&nbsp;&nbsp;&nbsp;0.0616&nbsp;&nbsp;0.0607
>
> **W3. Hyper-parameter sensitivity: The base value  seems to vary significantly across tasks (0.6 to 2.5) as per Appendix, suggesting the method may require careful tuning per dataset. Is there a way to tackle this?**
>
> Thank you for your valuable comment. We note that significant distributional differences exist across datasets, leading to corresponding variations in the NS score distributions of their training samples. This directly leads to the need to adjust the optimal parameter settings for different datasets. To investigate this matter empirically, we first evaluated the performance under different values of the base value $\sigma$ on the CIFAR-100 dataset using ResNet-110.
>
> As shown in the table below, setting $\sigma=0$ leads to significant performance degradation, primarily because the original NS scores of some samples are too small (close to zero), which hinders effective network optimization. The model achieves optimal performance when $\sigma$ is set to 0.8. Given that our study involves multiple tasks with substantial differences in dataset distributions, we recommend adopting a similar hyperparameter selection strategy to determine the optimal $\sigma$ value for different applications.
>
> | $\sigma$ | 0.0  | 0.1  | 0.5  |   0.8    | 1.0  | 1.5  | 1.8  |
> |:--------:|:----:|:----:|:----:|:--------:|:----:|:----:|:----:|
> | Acc (%)  | 67.5 | 70.9 | 71.3 | **72.1** | 71.7 | 71.5 | 71.4 |
>
> We fully understand your concern that the hyperparameter $\sigma$ needs to be tuned for different datasets. This is indeed a common issue shared by many classic methods, such as Focal Loss [1] and SCE [2]. It is worth noting that our method involves only a limited number of hyperparameters, resulting in a relatively low tuning cost. In the future, it would be valuable to explore adaptive mechanisms for parameters like $\sigma$ to ultimately achieve intervention-free hyperparameter tuning.
>
> *[1] Lin T Y, Goyal P, Girshick R, et al. Focal loss for dense object detection. ICCV 2017: 2980-2988.*
>
> *[2] Wang Y, Ma X, Chen Z, et al. Symmetric cross entropy for robust learning with noisy labels. ICCV 2019: 322-330.*

---

> ### Author Response · Authors · 2025-11-27
> **Official Comment by Authors (Part 4)**
>
> **Response to Questions**:
>
> ---
>
> **Q1. Do the authors have some idea how can they tackle the computation overhead limitation?**
>
> Please refer to the reply from **W1**.
>
> **Q2. How is the optimal group size determined across different tasks?Is group size a sensitive hyper-parameter? What principles guide the selection—does it depend on batch size, number of classes, or dataset characteristics?**
>
> Please refer to the reply from **W2**.
>
> **Q3. How does NS behave in the presence of outliers or anomalous samples? Based on the image classification results (Section 4.2), the Winner-Strengthening strategy performs better on ImageNet, which has label noise. Does favoring winners risk marginalizing legitimate hard samples or minority patterns that appear outlier-like?**
>
> Thank you for this insightful question. We acknowledge that datasets such as ImageNet contain certain anomalous and hard-to-recognize samples that can challenge model predictions. To examine whether the Winner-Strengthening strategy adversely affects the handling of hard samples, we conducted a dedicated evaluation on the ImageNet-Hard dataset [3]. This dataset comprises numerous images that are generally challenging for existing models to classify correctly, making it suitable for evaluating the robustness of methods on hard samples.
>
> Using ResNet-18 as the baseline, we performed experiments as summarized in the table below. It can be observed that after incorporating the Winner-Strengthening strategy, the model's performance (%) on hard samples did not exhibit significant degradation. This indicates that the proposed method poses a low risk of marginalizing legitimate hard samples or minority patterns.
>
> | Dataset       | ResNet-18 | w/ NS-WS |
> |:------------- |:---------:|:--------:|
> | ImageNet-Hard |    9.9    | **10.2** |
>
> *[3] Taesiri M R, Nguyen G, Habchi S, et al. Imagenet-hard: The hardest images remaining from a study of the power of zoom and spatial biases in image classification. NeurIPS 2023: 35878-35953.*
>
> **Q4. Do the authors believe this approach will transfer effectively to text-based or other non-vision tasks? The stitching mechanism is naturally suited to images, but how would competition be simulated for text (concatenation of sentences/tokens?)**
>
> Yes. We believe that the proposed method has general potential beyond visual tasks. To validate this, we conducted additional experiments on text classification tasks, using two public datasets (IMDB[4] and AG News [5]) and employing TextCNN [6] and BERT-base [7] as baseline models for evaluation. Specifically, for text-based samples, we concatenated four text instances into a single input, which was then converted into sequence or token representations. Subsequently, we computed the NS score for each text via model inference and applied the NS-LF strategy to adjust the training loss accordingly. The experimental results presented in the table below demonstrate that our method achieves consistent performance improvements in text classification tasks, indicating its cross-modal applicability.
>
> | Dataset | TextCNN | w/ NS-LF | BERT-base | w/ NS-LF |
> |:------- |:-------:|:--------:|:---------:|:--------:|
> | IMDB    |  75.6   | **76.6** |   91.8    | **92.2** |
> | AG News |  90.3   | **91.0** |   94.4    | **94.7** |
>
> *[4] Maas A, Daly R E, Pham P T, et al. Learning word vectors for sentiment analysis. ACL 2011: 142-150.*
>
> *[5] Zhang X, Zhao J, LeCun Y. Character-level convolutional networks for text classification. NeurIPS 2015: 649-657.*
>
> *[6] Kim Y. Convolutional Neural Networks for Sentence Classification. EMNLP 2014: 1746-1751.*
>
> *[7] Devlin J, Chang M W, Lee K, et al. Bert: Pre-training of deep bidirectional transformers for language understanding. NAACL 2019: 4171-4186.*

---

### Official Review · Reviewer_XAue · 2025-10-31

**Soundness:** 3
**Presentation:** 2
**Contribution:** 2
**Rating:** 4
**Confidence:** 2

**Summary:**

This paper introduces Darwinian Optimization, a bio-inspired training paradigm for deep neural networks based on the principle of natural selection. By explicitly modeling competition among samples through a Natural Selection (NS) score, the method dynamically adjusts per-sample loss weights to mimic ecological adaptation, enabling more balanced, efficient, and generalizable optimization of deep networks.

**Strengths:**

The key strength of this paper lies in introducing a biologically inspired optimization framework that embeds the principle of natural selection into deep network training. By modeling explicit inter-sample competition, it breaks away from uniform loss optimization, enabling adaptive and ecologically balanced learning that enhances both generalization and robustness. Moreover, the method is lightweight, architecture-agnostic, and easily integrable into existing training pipelines, demonstrating strong universality and scalability.

**Weaknesses:**

1.The analogy between biological evolution and sample competition remains conceptual without formal mathematical grounding.
2.The paper lacks detailed ablation studies isolating the effects of design components such as stitching, σ, and ρ.
3.Although the method achieves stable performance improvements across multiple datasets and network architectures, the gains appear to be relatively modest.
4.The “evolutionary rethinking” section repeats background rather than deepens analysis.
5.How should we interpret the phrase in the abstract: “thereby forming an optimization process that more closely mimics a Darwinian ecological equilibrium”?

**Questions:**

See the weakness section.

---

> ### Author Response · Authors · 2025-11-27
> **Official Comment by Authors (Part 1)**
>
> We sincerely appreciate your thoughtful feedback. We have carefully addressed each of your comments in the revised manuscript. Your suggestions have substantially improved the quality of our work, and we are grateful for your valuable input.
>
> **W1. The analogy between biological evolution and sample competition remains conceptual without formal mathematical grounding.**
>
> We sincerely thank the reviewer for this insightful comment. We fully agree that a formal mathematical foundation strengthens the contribution. In response, we have significantly revised the manuscript to ground the biological evolution analogy in a mathematical framework, rather than leaving it as a mere metaphor.
>
> Specifically, we first established a formal mapping between the key components of supervised learning and evolutionary concepts. Building upon this, we propose a formal proposition on the "Evolutionary-Learning Correspondence" and establish their structural correspondence through two aspects, i.e., the objective function duality and improvement dynamics. This work provides a mathematical foundation for interpreting network optimization from an evolutionary perspective, revealing the structural similarity between biological evolution and learning optimization mechanisms.
>
> Please refer to Section 3.1 of the revised manuscript for the complete updated content, with key modifications highlighted.
> > ''To establish a mathematical foundation for the evolutionary analogy, we define a mapping between the components of supervised learning and evolutionary concepts:
> > * **Population structure**: The training dataset $\mathcal{D}$ corresponds to a population of individuals, where each sample $(x_i, y_i)$ represents an individual.
> > * **Fitness function**: We define the fitness of an individual as $h_i(\theta) = M - \ell_i(\theta)$, where $M$ is a constant ensuring $f_i(\theta) > 0$. This suggests that high fitness is associated with low loss.
> > * **Selective pressure**: The gradient $\nabla_\theta \ell_i(\theta)$ provides directional pressure analogous to natural selection, driving the system toward fitter states.
> >
> > **Proposition 1 (Evolutionary-Learning Correspondence)**. Under the mapping $\phi$ defined by:
> > \begin{align*}
> >     \phi(\text{population}) = \mathcal{D}, \quad \phi(\text{individual } i)= (x_i, y_i), \quad
> >     \phi(\text{fitness of individual } i) = M - \ell_i(\theta), \quad \phi(\text{selective pressure}) = \nabla_\theta \ell_i(\theta),
> > \end{align*}
> > there exists a structural correspondence between the evolutionary process maximizing population fitness and the learning process minimizing empirical risk, characterized by equivalent optimization objectives and similar iterative improvement dynamics.
> >
> > We establish the structural correspondence through two key aspects:
> >
> > 1) **Objective function duality**. The evolutionary objective of maximizing average fitness:
> > \begin{equation}
> >    \max_{\theta} \frac{1}{N} \sum_{i=1}^N h_i(\theta) = \max_{\theta} \frac{1}{N} \sum_{i=1}^N [M - \ell_i(\theta)]
> > \end{equation}
> > is mathematically equivalent to the learning objective of minimizing empirical risk:
> > \begin{equation}
> >    \min_{\theta} \frac{1}{N} \sum_{i=1}^N \ell_i(\theta)
> > \end{equation}
> > since maximizing $M - \ell_i(\theta)$ is equivalent to minimizing $\ell_i(\theta)$ for constant $M$.
> >
> > 2) **Improvement dynamics**. Both processes employ iterative improvement strategies: In evolution, fitter individuals are more likely to reproduce, gradually improving population fitness; In learning, parameters are updated in the direction that reduces loss: $\theta_{t+1} = \theta_t - \eta \cdot \nabla_\theta L(\theta)$. While their update process differ, both mechanisms drive the system toward better performance over time.
> >
> > This formal correspondence provides a mathematical foundation for interpreting network optimization through an evolutionary lens, highlighting the structural similarities between biological evolution and learning optimization.''

---

> ### Author Response · Authors · 2025-11-27
> **Official Comment by Authors (Part 2)**
>
> **W2. The paper lacks detailed ablation studies isolating the effects of design components such as stitching, σ, and ρ.**
>
> Thank you for your constructive feedback. As suggested, we have conducted supplementary experiments on the CIFAR-100 dataset using ResNet-110. These experimental results and analyses have been incorporated into Appendix F of the revised manuscript.
>
> **First**, we evaluated five different grouped stitching configurations, including 1×2 (2 images), 2×2 (4 images), 2×4 (8 images), 4×2 (8 images), and 4×4 (16 images) layouts. As shown in the table below, our evaluation revealed that group sizes of 2 and 4 both achieved better and comparable performance. Therefore, we adopted the 2/4-group configuration across all experiments.
>
> | Group   | 1x2  |   2x2    | 2x4  | 4x2  | 4x4  |
> |:------- |:----:|:--------:|:----:|:----:|:----:|
> | Acc (%) | 72.0 | **72.1** | 71.7 | 71.4 | 71.7 |
>
> **Second**, we evaluated the base value $\sigma$. As shown in the table below, setting $\sigma=0$ leads to significant performance degradation, primarily because the original NS scores of some samples are too small (close to zero), which hinders effective network optimization. The model achieves optimal performance when $\sigma$ is set to 0.8. Given that our study involves multiple tasks with substantial differences in dataset distributions, we recommend adopting a similar hyperparameter selection strategy to determine the optimal $\sigma$ value for different applications.
>
> | $\sigma$       | 0.0  | 0.1  | 0.5  |   0.8    | 1.0  | 1.5  | 1.8  |
> | ------- |:----:|:----:|:----:|:--------:|:----:|:----:|:----:|
> | Acc (%) | 67.5 | 70.9 | 71.3 | **72.1** | 71.7 | 71.5 | 71.4 |
>
> **Third**, on the CIFAR-100 dataset, we adopted the winner-strengthening strategy with the parameter $\rho$ set to 1. To validate the rationality of this parameter choice, we further evaluated the model performance with different values of $\rho$. As shown in the table below, the model achieves optimal performance when $\rho=1$. Based on this empirical result, we set $\rho$ at 1 in our experiments without further fine-tuning of this parameter.
>
> | $\rho$       | 0.0  | 0.1  | 0.5  | 0.8  |   1.0    | 1.5  | 1.8  |
> | ------- |:----:|:----:|:----:|:----:|:--------:|:----:|:----:|
> | Acc (%) | 70.3 | 71.3 | 71.6 | 71.8 | **72.1** | 71.9 | 71.8 |
>
>
> **W3. Although the method achieves stable performance improvements across multiple datasets and network architectures, the gains appear to be relatively modest.**
>
> Thank you for this meaningful comment. We sincerely acknowledge that the proposed method does not achieve significant improvements across all tasks and backbone networks. However, it is worth noting that achieving consistent improvements across diverse tasks and architectures remains challenging, even for standard classification benchmarks.
>
> Taking ImageNet 1K as an example, we compiled the top 1 accuracy of several models with comparable parameter counts published between 2021 and 2025 on this dataset (as shown in the table below, sourced from a recent paper [1]). It can be observed that the year-over-year improvements on this benchmark are generally modest. Given the scale and difficulty of ImageNet 1K, even small gains are considered meaningful.
>
> | Method          | Venue        | Param (M) | ACC (%) |
> |:--------------- |:------------ |:---------:|:-------:|
> | Swin-S          | ICCV 2021    |    50     |  83.0   |
> | CoAtNet-1       | NeurIPS 2021 |    42     |  83.3   |
> | Focal-Small     | NeurIPS 2021 |    51     |  83.5   |
> | ConvNeXt-S      | CVPR 2022    |    50     |  83.1   |
> | CSWin-S         | CVPR 2022    |    35     |  83.6   |
> | MViTv2-S        | CVPR 2022    |    35     |  83.6   |
> | UniFormer-B     | TPAMI 2023   |    50     |  83.9   |
> | SG-Former-M     | ICCV 2023    |    39     |  84.1   |
> | InternImage-S   | CVPR 2023    |    50     |  84.2   |
> | ConvFormer-S36  | TPAMI 2024   |    40     |  84.1   |
> | MogaNet-B       | ICLR 2024    |    44     |  84.3   |
> | TransNeXt-Small | CVPR 2024    |    50     |  84.7   |
> | VMamba-S        | NeurIPS 2024 |    44     |  83.5   |
> | VMambaV3-S*     | NeurIPS 2024 |    50     |  83.6   |
> | LocalVMamba-S   | ECCV 2024    |    50     |  83.7   |
> | MambaOut-Small  | CVPR 2025    |    48     |  84.1   |
>
> Furthermore, we aggregated results across the 12 public datasets and a total of 69 baseline models reported in Tables 1 to 5 of our paper. The statistics show that our method achieves an improvement of at least 0.5% in 52 out of 69 cases (75%), and an improvement of at least 1% in 34 out of 69 cases (49%). These results demonstrate the stable effectiveness and broad applicability of the proposed method across various settings.
>
> *[1] Yu W, Wang X. Mambaout: Do we really need mamba for vision? CVPR 2025: 4484-4496.*

---

> ### Author Response · Authors · 2025-11-27
> **Official Comment by Authors (Part 3)**
>
> **W4. The “evolutionary rethinking” section repeats background rather than deepens analysis.**
>
> Thank you for your important suggestion. To enhance the theoretical rigor and analytical depth of this section, we have made substantial additions and reorganized the content.
>
> **First**, we established a formal mapping between the fundamental components of supervised learning and evolutionary concepts. Based on this, we proposed a formal "Evolution-Learning Correspondence" proposition and developed the structural relationship between the two fields from the perspectives of optimization objective duality and dynamic processes. This provides a mathematical foundation for interpreting network optimization from an evolutionary perspective (see W1 response for details).
>
> **Second**, we re-examined the classical supervised learning paradigm through Empirical Risk Minimization, identifying its core limitation as the homogeneous treatment resulting from the uniform weighting scheme (i.e., w_i $\equiv$ 1). This prevents the learning process from developing dynamic selection pressures based on individual fitness, analogous to those found in ecosystems. The detailed analysis can be found in Section 3.1 of the revised manuscript.
>
> We believe these additions significantly enhance the theoretical depth and rigor of our study.
>
> > By establishing an analogy between network optimization and ecosystem evolution, we can critically re-examine the limitations of the classical supervised learning paradigm with Empirical Risk Minimization (ERM). The standard ERM framework minimizes the average loss over the training dataset: $\min_\theta \frac{1}{N}\sum_{i=1}^N \ell(x_i, y_i; \theta)$, which implicitly assigns equal weight $w_i = 1$ to every sample $(x_i, y_i)$. This uniform weighting scheme, while statistically well-motivated, fails to account for the competitive dynamics among samples. In ecological terms, the ERM objective applies a constant selective pressure $\nabla_\theta \ell(x_i, y_i; \theta)$ to all individuals, regardless of their fitness or the population composition. The gradient signal that drives optimization, $\frac{1}{N}\sum_{i=1}^N \nabla_\theta \ell(x_i, y_i; \theta)$, represents a global average that lacks the fine-grained adaptation to individual competitive advantages.
> >
> > The core limitation lies in the homogeneity of the weighting function $w_i \equiv 1$, which prevents the emergence of dynamic selection pressures analogous to those in natural ecosystems. Consequently, the network optimization process cannot prioritize samples with higher fitness or protect developing minority patterns $\mathcal{D} _ {\text{min}}$ from being overwhelmed by dominant patterns $\mathcal{D}_{\text{maj}}$. From an evolutionary perspective, this eliminates the natural selection mechanism essential for maintaining diversity and adapting to complex environments, often leading to premature convergence and limited generalization capability.
>
> **W5. How should we interpret the phrase in the abstract: “thereby forming an optimization process that more closely mimics a Darwinian ecological equilibrium”?**
>
> We sincerely appreciate your valuable question. Considering that the biological analogy used in the original abstract was overly metaphorical in that context and did not accurately convey the technical content, we have removed that analogy in the revised manuscript and replaced it with a direct description of the algorithmic mechanism.
>
> The revised and complete sentence is as follows:
> > "This score is further used to dynamically adjust the loss weight of each sample, facilitating an adaptive network optimization process driven by competitive interactions among training samples."
>
> This revision more precisely reflects the essence of our method: by quantifying the competitive relationships among samples, the network optimization process is dynamically adjusted. We believe the modified statement significantly enhances the technical precision and academic rigor of the abstract.

---

### Official Review · Reviewer_MYGf · 2025-11-01

**Soundness:** 3
**Presentation:** 3
**Contribution:** 3
**Rating:** 4
**Confidence:** 4

**Summary:**

The paper proposes Natural Selection (NS) as a novel optimization method, drawing inspiration from species competition and adaptation in natural ecosystems. NS introduces a dynamic competition mechanism among training samples. This bio-inspired approach is designed to mitigate classic deep learning training challenges, including class imbalance bias, insufficient learning of hard samples, and instability due to noisy data by applying non-uniform selective pressure.

**Strengths:**

- The paper is well-written, and its central perspective of connecting Darwinian theory of species to deep learning provides a novel perspective.
- The experiments cover a wide range of factors, including diverse architectures, multiple datasets, class imbalance and domain adaptation scenarios.

- The approach shows strong performance and maintains its efficacy even under challenging class imbalance conditions.

**Weaknesses:**

- The average empirical improvements reported in Tables 1, 2, and 3 for NS are marginal especially in the large-scale data like Imagenet 1K.

- Furthermore, the number of experimental seeds used is not specified, making the reported gains difficult to interpret. Statistical significance analysis is needed to confirm that these marginal improvements are meaningful and not due to random variation.

-  Interpreting the results is difficult because the authors label their overall method as "NS" while the actual techniques applied in the experiments are "Winner Strengthening" or "Loser Focussing are not specified clearly.

- The abstract description of the core mechanism ("stitching and scaling a group of samples") is vague. Without explicit mathematical details, it is difficult to assess the computational complexity. Provide a detailed comparison of the per-iteration computational overhead of NS versus comparisons.

- The current set of empirical comparisons is heavily biased toward loss function modification methods (e.g., Focal Loss, GCE, PolyLoss). The evaluation lacks crucial baselines from the highly effective data-level rebalancing category, such as those based on sampling. This gap makes it impossible to assess if the proposed method's gains are simply equivalent to or outperformed by simpler, established sampling techniques.

**Questions:**

See Weakness

---

> ### Author Response · Authors · 2025-11-27
> **Official Comment by Authors (Part 1)**
>
> We sincerely appreciate your valuable suggestions and insightful questions, which have provided important guidance for improving our work. We have carefully addressed each comment with corresponding revisions, as detailed in our responses below.
>
> **W1. The average empirical improvements reported in Tables 1, 2, and 3 for NS are marginal especially in the large-scale data like Imagenet 1K.**
>
> Thank you for your important feedback. We sincerely acknowledge that the proposed method does not achieve significant improvements across all tasks and backbone networks. However, it is worth noting that achieving consistent improvements across diverse tasks and architectures remains challenging, even for standard classification benchmarks.
>
> Taking ImageNet 1K as an example, we compiled the top 1 accuracy of several models with comparable parameter counts published between 2021 and 2025 on this dataset (as shown in the table below, sourced from a recent paper [1]). It can be observed that the year-over-year improvements on this benchmark are generally modest. Given the scale and difficulty of ImageNet 1K, even small gains are considered meaningful.
>
> | Method| Venue        | Param (M) | ACC (%) |
> |:--------------- |:------------ |:---------:|:-------:|
> | Swin-S| ICCV 2021|    50     |  83.0   |
> | CoAtNet-1| NeurIPS 2021|    42     |  83.3   |
> | Focal-Small| NeurIPS 2021|    51     |  83.5   |
> | ConvNeXt-S| CVPR 2022|    50     |  83.1   |
> | CSWin-S| CVPR 2022|    35     |  83.6   |
> | MViTv2-S| CVPR 2022|    35     |  83.6   |
> | UniFormer-B| TPAMI 2023|    50     |  83.9   |
> | SG-Former-M| ICCV 2023|    39     |  84.1   |
> | InternImage-S| CVPR 2023|    50     |  84.2   |
> | ConvFormer-S36| TPAMI 2024|    40     |  84.1   |
> | MogaNet-B| ICLR 2024|    44     |  84.3   |
> | TransNeXt-Small| CVPR 2024|    50     |  84.7   |
> | VMamba-S| NeurIPS 2024|    44     |  83.5   |
> | VMambaV3-S*| NeurIPS 2024|    50     |  83.6   |
> | LocalVMamba-S| ECCV 2024|    50     |  83.7   |
> | MambaOut-Small| CVPR 2025|    48     |  84.1   |
>
> Furthermore, we aggregated results across the 12 public datasets and a total of 69 baseline models reported in Tables 1 to 5 of our paper. The statistics show that our method achieves an improvement of at least 0.5% in 52 out of 69 cases (75%), and an improvement of at least 1% in 34 out of 69 cases (49%). These results demonstrate the stable effectiveness and broad applicability of the proposed method across various settings.
>
> *[1] Yu W, Wang X. Mambaout: Do we really need mamba for vision? CVPR 2025: 4484-4496.*
>
> **W2. Furthermore, the number of experimental seeds used is not specified, making the reported gains difficult to interpret. Statistical significance analysis is needed to confirm that these marginal improvements are meaningful and not due to random variation.**
>
> Thank you for your valuable comment. In our experiments, we performed multiple runs using three random seeds (2024, 2025, 2026) and reported the average performance. We systematically compiled the mean results and standard deviations across the 12 public datasets considered in this study, as shown in the table below. Overall, the performance gains achieved by our method are significantly greater than random fluctuations, indicating that the observed improvements are statistically significant. We have added these statistical results to the revised version of the manuscript (see Table 6). Thank you again for your valuable suggestions.
>
> | Datasets    | mean±std  | Datasets   | mean±std  | Datasets  | mean±std  | Datasets     | mean±std  |
> |:----------- |:---------:|:---------- |:---------:|:--------- |:---------:|:------------ |:---------:|
> | ImageNet-1K | 73.9±0.05 | Twitter I  | 77.0±0.45 | Instagram | 83.7±0.21 | Office-Home  | 73.0±0.19 |
> | CIFAR-10    | 91.8±0.08 | Twitter II | 73.3±0.34 | FI        | 64.2±0.26 | CIFAR-LT-10  | 74.9±0.43 |
> | CIFAR-100   | 70.9±0.22 | Flickr     | 85.5±0.11 | EmoSet    | 75.1±0.12 | CIFAR-LT-100 | 43.9±0.18 |
>
> **W3. Interpreting the results is difficult because the authors label their overall method as "NS" while the actual techniques applied in the experiments are "Winner Strengthening" or "Loser Focussing are not specified clearly**
>
> Thank you for pointing out this important issue of clarity. Following your suggestion, we have revised Section 3.3 to formally define the two strategies as "NS-based Winner-Strengthening (NS-WS)" and "NS-based Loser-Focusing (NS-LF)", emphasizing that NS serves as a general framework while NS-WS and NS-LF represent two specific instantiations.
>
> On this basis, we have updated all relevant tables and analyses in the experimental section to clearly indicate which specific strategy (NS-WS or NS-LF) each result corresponds to, thereby avoiding potential ambiguity from using "NS" alone. We believe these revisions significantly improve the interpretability of results. The corresponding changes have been highlighted on pages 5–9 of the revised manuscript for your reference.

---

> ### Author Response · Authors · 2025-11-27
> **Official Comment by Authors (Part 2)**
>
> **W4. The abstract description of the core mechanism ("stitching and scaling a group of samples") is vague. Without explicit mathematical details, it is difficult to assess the computational complexity. Provide a detailed comparison of the per-iteration computational overhead of NS versus comparisons.**
>
> Thank you for pointing this out. Following your suggestion, we have revised the relevant statements in the abstract to provide a clearer explanation of the proposed method. The revised wording is as follows:
>
> > ''NS introduces a competition mechanism by first assembling a group of samples into a composite image and then downscaling it to the original input size for model inference. Each sample is then assigned a natural selection score based on the model's predictions on this composite image, reflecting its competitive status within the group.''
>
> This revision specifies the procedural details of our method and we have incorporated this change into the revised version of the manuscript.
>
> **In addition**, your comment regarding the lack of explicit mathematical details is crucial for enhancing the rigor of the paper. As per your guidance, we have added a formal mathematical description of the image stitching and scaling operations in Section 3.2 of the revised manuscript, as detailed below:
>
> > ''As illustrated in Fig. 1(b), given a set of training samples $\mathcal{G}=\{(x_i,y_i)\}_{i=1}^m$ (e.g., $m=4$), where each sample has a spatial size of $H_0 \times W_0 \times 3$, we first arrange the samples into an $R \times C$ grid layout (with $R \times C = m$) and concatenate them along spatial dimensions to form a composite image:
> \begin{equation}
>     S = \text{Stitch}(x_1, x_2, \dots, x_m) \in \mathbb{R}^{R \cdot H_0 \times C \cdot W_0 \times 3}.
> \end{equation}
>
> > The composite image is subsequently rescaled to the original input size via bilinear interpolation, producing $S' \in \mathbb{R}^{H_0 \times W_0 \times 3}$, which is then passed to the classifier $f_\theta(\cdot)$ to obtain the output logits $z = f_\theta(S') \in \mathbb{R}^K$.''
>
> This addition presents the process using mathematical notation, clarifying the specific computations for stitching layout and scale transformation. Furthermore, the PyTorch implementation provided in Appendix A complements the mathematical description presented above, enabling readers to gain a more intuitive understanding of the methodological details.
>
> **Finally**, in addition to the overall computational cost comparison provided in Appendix Table 12, we provide a per-epoch timing analysis (see Table 13) measured on an NVIDIA RTX 4090 GPU. This analysis evaluates the two core NS steps, image stitching with scaling and model inference, with detailed results shown in the table below.
>
> | Step                        | CIFAR-10 | CIFAR-100 |
> |:--------------------------- |:--------:|:---------:|
> | Image stitching and scaling |  0.14 s  |  0.14 s   |
> | Model inference             |  2.31 s  |  2.87 s   |
>
> This analysis reveals that the additional computational cost primarily originates from the model inference step, while the time costs of image stitching and scaling are minimal, resulting in a total computational cost that remains within a reasonable range.

---

> ### Author Response · Authors · 2025-11-27
> **Official Comment by Authors (Part 3)**
>
> **W5. The current set of empirical comparisons is heavily biased toward loss function modification methods (e.g., Focal Loss, GCE, PolyLoss). The evaluation lacks crucial baselines from the highly effective data-level rebalancing category, such as those based on sampling. This gap makes it impossible to assess if the proposed method's gains are simply equivalent to or outperformed by simpler, established sampling techniques.**
>
> Thank you for this meanful suggestion. We have selected the following three classic sampling-based data balancing methods for comparative analysis:
>
> * **Class-Balanced Sampling (CBS)**. This method aims to make the expected sampling probability equal across classes. A typical implementation uses a two-stage strategy: first uniformly sampling a class from the class set, and then uniformly selecting a sample from that class. Equivalently, it can be viewed as assigning a sampling weight to each sample from class $k$ that is inversely proportional to its class frequency $n_k$.
>
> * **Square-Root Sampling (SRS)**. As a common variant of class-balanced sampling, this method applies a square-root transformation to class frequencies to smooth sampling bias. Concretely, the class-level sampling probability is proportional to $\sqrt{n_k}$, and with the two-stage implementation, the instance-level probability becomes proportional to $\frac{1}{\sqrt{n_k}}$. This achieves a trade-off between frequency-sensitive and class-balanced sampling.
>
> * **Progressively-Balanced Sampling (PBS)**. This method performs dynamic interpolation between frequency-based sampling and class-balanced sampling. The sampling probability is a weighted combination of them, with the weight of class-balanced sampling gradually increasing during the training process to achieve a smooth transition from frequency-preference to class balance.
>
> We have evaluated the three sampling methods on the Office-Home and CIFAR-LT-10/100 datasets, with results summarized in the table below. Experimental results indicate that sampling-based methods yield no notable improvement on the SFDA task, and while exhibiting certain gains under some long-tailed classification settings, their performance remains inconsistent. In contrast, our method consistently outperforms all sampling baselines on both types of tasks, confirming its effectiveness and competitive advantage. We have incorporated the corresponding results into the revised manuscript (see Tables 4-5). We sincerely appreciate your suggestion, which has been instrumental in improving the completeness of this study.
>
> | Method   |  Ar→Cl   |  Ar→Pr   |  Ar→Rw   |  Cl→Ar   |  Cl→Pr   |  Cl→Rw   |  Pr→Ar   |  Pr→Cl   |  Pr→Rw   |  Rw→Ar   |  Rw→Cl   |  Rw→Pr   |   Avg.   |
> |:-------- |:--------:|:--------:|:--------:|:--------:|:--------:|:--------:|:--------:|:--------:|:--------:|:--------:|:--------:|:--------:|:--------:|
> | Baseline |   58.0   |   78.8   |   81.9   |   69.4   |   79.7   |   78.7   |   68.2   |   55.0   |   81.9   |   73.8   |   58.7   |   83.8   |   72.3   |
> | CBS      |   57.9   |   78.3   |   81.7   |   68.7   |   79.6   |   78.3   |   68.3   |   55.4   |   82.1   |   72.8   |   59.4   |   84.2   |   72.2   |
> | SRS      |   57.4   |   78.7   |   81.6   |   68.4   |   79.5   |   78.5   |   68.0   |   56.1   |   81.9   |   73.5   |   58.4   |   84.5   |   72.2   |
> | PBS      |   57.6   |   78.5   |   82.2   |   68.4   |   79.7   |   78.6   |   68.4   |   56.3   |   82.2   |   73.5   |   59.5   |   83.9   |   72.4   |
> | **NS**   | **59.1** | **79.1** | **82.4** | **70.0** | **79.8** | **78.8** | **68.8** | **57.0** | **82.3** | **73.9** | **60.1** | **85.0** | **73.0** |
>
> &nbsp;&nbsp;&nbsp;&nbsp;&nbsp;&nbsp;&nbsp;&nbsp;&nbsp;&nbsp;&nbsp;&nbsp;&nbsp;&nbsp;&nbsp;&nbsp;&nbsp;&nbsp;&nbsp;&nbsp;&nbsp;&nbsp;&nbsp; **CIFAR-LT-10**&nbsp;&nbsp;&nbsp;&nbsp;&nbsp;&nbsp;&nbsp;&nbsp;&nbsp;&nbsp;&nbsp;&nbsp;&nbsp;&nbsp;&nbsp;  **CIFAR-LT-100**
> | Method   |   200    |   100    |    50    |    10    |   200    |   100    |    50    |    10    |
> |:-------- |:--------:|:--------:|:--------:|:--------:|:--------:|:--------:|:--------:|:--------:|
> | Baseline |   64.0   |   69.5   |   72.7   |   86.7   |   33.6   |   38.4   |   42.1   |   53.8   |
> | CBS      |   61.0   |   68.5   |   75.1   |   86.6   |   26.8   |   30.5   |   38.5   |   53.9   |
> | SRS      |   61.6   |   69.9   |   75.2   |   86.5   |   32.2   |   37.4   |   41.6   |   55.1   |
> | PBS      |   63.6   |   68.7   |   75.2   |   86.8   |   28.7   |   31.4   |   38.3   |   54.6   |
> | **NS**   | **65.9** | **71.2** | **75.3** | **87.3** | **35.4** | **39.7** | **43.9** | **56.4** |

---

### Author Response · Authors · 2025-12-02
**General Response**

Dear Area Chair and Reviewers:

We sincerely thank the reviewers for their valuable insights and constructive feedback, which have significantly improved our work. We have thoroughly addressed all the points raised and incorporated the corresponding revisions, which have been highlighted in the manuscript.

The **core contribution** of this paper is a biologically-inspired Natural Selection method that introduces inter-sample competition during training. By computing a natural selection score and adaptively adjusting loss weights accordingly, our method promotes more effective network optimization. Extensive experiments on 12 public datasets across four vision tasks, involving 69 baseline models, demonstrate its stable performance gains and broad applicability. The method offers a new perspective on deep network optimization, characterized by easy integration and strong potential for generalization across domains.

The reviewers broadly recognized this paper's novelty, significant contributions, and broad applicability. Specific **positive feedback** includes:
* Reviewers MYGf and Bfq8 highlighted the high originality of the Darwin-inspired biological approach;
* Reviewers MYGf, XAue, and Bfq8 acknowledged the generality and broad applicability;
* Reviewers MYGf and Bfq8 commended the clear writing and comprehensive experiments;
* Reviewer Bfq8 specifically praised the interesting work, clear visualizations, and appropriate analogy.

Reviewer concerns primarily stem from clarifiable misunderstandings and addressable limitations, which we have thoroughly resolved in our responses and revisions. **Key points** include:
* Method performance gains required further validation (Reviewers MYGf, XAue) -> Addressed with additional statistical results in the rebuttal;
* Experiments needed stronger statistical and ablative analysis (Reviewers MYGf, XAue) -> Addressed by adding Tables 6-9 and analysis in Appendices E, F;
* Comparisons with sampling-based methods were suggested (Reviewer MYGf) -> Addressed by including three classical sampling methods in Tables 4, 5;
* Certain descriptions lacked clarity (Reviewers MYGf, XAue) -> Addressed through comprehensive revisions to relevant sections;
* Biological analogy needed mathematical grounding (Reviewer XAue) -> Addressed by adding mathematical formulations and theoretical analysis in Sections 3.1, 3.2;
* Questions on computational cost, group size, and hyperparameter sensitivity (Reviewer Bfq8) -> Addressed with supplementary experiments in the rebuttal and data in Appendix F, I Tables 7, 8, 13;
* Inquiries about artificial edge patterns and backpropagation (Reviewer Bfq8) -> Addressed with detailed experimental analysis in the rebuttal;
* Curiosity about performance on hard samples and text-based tasks (Reviewer Bfq8) -> Addressed with supplementary results in Appendix H Tables 10, 11.

Once again, we extend our sincere gratitude for your time and insightful comments.

Best regards,

The Authors

---

### Meta-Review · Area_Chair_F16J · 2026-01-12

**Summary:**

The main concerns were:
- improvements that look modest on large-scale benchmarks (especially ImageNet-1K) and initially lacked statistical backing
- missing comparisons to effective sampling-based rebalancing methods, making it hard to judge whether NS is actually better than simpler data-level baselines
- the biological analogy and “ecological equilibrium” narrative initially felt more metaphorical than formal, and ablations (stitching / σ / ρ) were insufficiently isolated
- questions about added computation (reported ~10–13% overhead) and sensitivity to design choices like group size.

Overall the rebuttal does meaningfully strengthen the paper (seeds/stats, added baselines, added analyses), and the method is simple enough that incremental gains across many settings are still useful. My remaining hesitation is mainly about novelty/necessity (is stitching competition truly giving something beyond standard confidence-based reweighting?) and whether the 'Darwinian' framing oversells what is essentially a practical heuristic.

**Reviewer Concerns:**

Addressed:

- ran 3 seeds and added mean±std summaries across datasets, arguing gains exceed random fluctuations.
- added comparisons against classical sampling approaches (CBS/SRS/PBS)
- expanded ablations (σ/ρ/stitching)
- they propose 'compute NS every k epochs' and show a speed/accuracy tradeoff table where k=4 keeps most gains with much lower overhead.
- added ImageNet-Hard results suggesting no obvious degradation on hard samples and a text-classification adaptation with small but consistent gains.

Still outstanding:

- Even with the revised framing, it remains unclear whether the stitched “competition” mechanism provides a principled advantage over simpler per-sample confidence weighting or established reweighting/meta-reweighting methods.

- Addresses edge artifacts and tries variants, but the why is still a bit hand-wavy (what property of the composite inference makes NS scores better than single-image confidence?).

- The paper/rebuttal should be cleaned up so ρ/σ roles and sensitivities are unambiguous and consistent (right now it’s easy to interpret ρ as a sign/polarity in one place and a tuned magnitude in another).

- The k-epoch trick is promising, but it changes the algorithm (stale NS scores).

**Reviewer Scores:**

- MYGf: 4 to likely 6
- XAue: 4 -> likely the same
- Bfq8: 8 -> stays 8

---

### Decision · Program_Chairs · 2026-01-26

Reject